# Accommodating Picky Customers: Regret Bound and Exploration Complexity for Multi-Objective Reinforcement Learning

**Jingfeng Wu**
Department of Computer Science
Johns Hopkins University
Baltimore, MD 21218
uuujf@jhu.edu

**Vladimir Braverman**
Department of Computer Science
Johns Hopkins University
Baltimore, MD 21218
vova@cs.jhu.edu

**Lin F. Yang**
Department of Electrical and Computer Engineering
University of California, Los Angeles
Los Angeles, CA 90095
linyang@ee.ucla.edu

## Abstract

In this paper we consider multi-objective reinforcement learning where the objectives are balanced using preferences. In practice, the preferences are often given in an adversarial manner, e.g., customers can be picky in many applications. We formalize this problem as an episodic learning problem on a Markov decision process, where transitions are unknown and a reward function is the inner product of a preference vector with pre-specified multi-objective reward functions. We consider two settings. In the online setting, the agent receives a (adversarial) preference every episode and proposes policies to interact with the environment. We provide a model-based algorithm that achieves a nearly minimax optimal regret bound $\widetilde{\mathcal{O}}\big(\sqrt{\min\{d, S\} \cdot H^2 SAK}\big)$, where $d$ is the number of objectives, $S$ is the number of states, $A$ is the number of actions, $H$ is the length of the horizon, and $K$ is the number of episodes. Furthermore, we consider preference-free exploration, i.e., the agent first interacts with the environment without specifying any preference and then is able to accommodate arbitrary preference vector up to $\epsilon$ error. Our proposed algorithm is provably efficient with a nearly optimal trajectory complexity $\widetilde{\mathcal{O}}\big(\min\{d, S\} \cdot H^3 SA/\epsilon^2\big)$. This result partly resolves an open problem raised by Jin et al. [2020].

## 1 Introduction

In single-objective reinforcement learning (RL), a *scalar reward* is pre-specified and an agent learns a policy to maximize the long-term cumulative reward [Sutton and Barto, 2018]. However, in many real-world applications, we need to optimize multiple objectives for the same (unknown) environment, even when these objectives are possibly contradicting [Roijers et al., 2013]. For example, in an autonomous driving application, each passenger may have a different preference of driving styles: some of the passengers prefer a very steady riding experience while other passengers enjoy the fast acceleration of the car. Therefore, traditional single-objective RL approach may fail to be applied in such scenarios. One way to tackle this issue is the *multi-objective reinforcement learning* (MORL) [Roijers et al., 2013, Yang et al., 2019, Natarajan and Tadepalli, 2005, Abels et al., 2018]

35th Conference on Neural Information Processing Systems (NeurIPS 2021).

method, which models the multiple objectives by a *vectorized reward*, and an additional *preference vector* to specify the relative importance of each objective. The agent of MORL needs to find policies to optimize the cumulative preference-weighted rewards.

If the preference vector is fixed or drawn from a fixed distribution, MORL is no more challenging than single-objective RL, as we can predict the objective to be optimized and apply (variants of) single-objective RL algorithms (e.g., [Azar et al., 2017]). However, more often in practice, the preference vector (under which the weighted objective needs to be optimized) is:

  (i) adversarially provided,
  (ii) or even not available in the learning phase.

Once more taking the autonomous driving application as example: (i) the intelligent system needs to adapt driving configurations to accommodate every customer, even though the next customer can have picky preference that is unpredictable; (ii) when the intelligent system is under development, the future customer cannot be known in advance, nonetheless, when the system is deployed, it has to be capable to accommodate any potential customer who rides the car. Such MORL examples are common, to name a few beyond the autonomous driving one: medical treatment must take care of every patient even in very rare health conditions; an education system should accommodate every student according to his/her own characteristics; and emergency response systems have to be responsible in all extreme cases. Due to these exclusive challenges that have not appeared in single-objective RL, new *sample-efficient* algorithms for MORL need to be developed, as well as their *theoretical grounds* need to be established.

In this work, we study *provable* sample-efficient algorithms for MORL that resolve the aforementioned issues. Specifically, we consider MORL on a finite-horizon Markov decision process (MDP) with an unknown transition kernel, $S$ states, $A$ actions, $H$ steps, and $d$ reward functions that represent the $d$ difference objectives. We investigate MORL problems in two paradigms: (i) *online MORL* (where the preferences are adversarially presented), and (ii) *preference-free exploration* (where the preferences are not available in the learning/exploration phase). The two settings and our contributions are explained respectively in the following.

**Setting (i): Online MORL.** We first consider an online learning problem to capture the challenge that preference vectors could be adversarially provided in MORL. In the beginning of each episode, the MORL agent is provided (potentially adversarially) with a preference vector, and the agent interacts with the unknown environment to collect data and rewards (that is specified by the provided preference). The performance of an algorithm is measured by the *regret*, i.e., the difference between the rewards collected by the agent and those would be collected by a theoretically optimal agent (who could use varying policies that adapt to the preferences). This setting generalizes the classical online single-objective RL problem [Azar et al., 2017] (where $d = 1$).

**Contribution (i).** For online MORL, we provide a provably efficient algorithm with a regret upper bound $\widetilde{\mathcal{O}}\big(\sqrt{\min\{d, S\} \cdot H^2 SAK}\big)$[1], where $K$ is the number of episodes for interacting with the environment. Furthermore, we show an information-theoretic lower bound $\Omega\big(\sqrt{\min\{d, S\} \cdot H^2 SAK}\big)$ for online MORL. These bounds together show that, ignoring logarithmic factors, our algorithm resolves the online MORL problems optimally.

**Setting (ii): Preference-Free Exploration.** We further consider an unsupervised MORL problem to capture the issue that preferences could be hard to obtain in the training phase. The MORL agent first interacts with the unknown environment in the absence of preference vectors; afterwards, the agent is required to use the collected data to compute near-optimal policies for an arbitrarily specified preference vector. The performance is measured by the *sample complexity*, i.e., the minimum amount of trajectories that an MORL agent needs to collect during exploration in order to be near-optimal during planning. This setting extends the recent proposed *reward-free exploration* problem [Jin et al., 2020] to MORL.

**Contribution (ii).** For preference-free exploration, we show that a simple variant of the proposed online algorithm can achieve nearly optimal sample complexity. In particular, the algorithm achieves a sample complexity upper bound $\widetilde{\mathcal{O}}\big(\min\{d, S\} \cdot H^3 SA/\epsilon^2\big)$ where $\epsilon$ is the tolerance of the planning error; and we also show a sample complexity lower bound, $\Omega\big(\min\{d, S\} \cdot H^2 SA/\epsilon^2\big)$, for any

---

[1] We use $\widetilde{\mathcal{O}}(\cdot)$ to hide (potential) polylogarithmic factors in $\mathcal{O}(\cdot)$, i.e., $\widetilde{\mathcal{O}}(n) := \mathcal{O}(n \log^k n)$ for sufficiently large $n$ and some absolute constant $k > 0$.

algorithm. These bounds suggest that our algorithm is optimal in terms of $d$, $S$, $A$, $\epsilon$ up to logarithmic factors. It is also worth noting that our results for preference-free exploration partly answer an open question raised by Jin et al. [2020]: reward-free RL is easier when the unknown reward functions enjoy low-dimensional representations (as in MORL).

**Paper Layout.** The remaining paper is organized as follows: the preliminaries are summarized in Section 2; then in Section 3, we formally introduce the problem of online MORL, our algorithm and its regret upper and lower bounds; then in Section 4, we turn to study the preference-free exploration problem in MORL, where we present an exploration algorithm with sample complexity analysis (with both upper bound and lower bound), and compare our results with existing results for related problems; finally, the related literature is reviewed in Section 5 and the paper is concluded by Section 6.

## 2 Preliminaries

We specify a finite-horizon Markov decision process (MDP) by a tuple of $(\mathcal{S}, \mathcal{A}, H, \mathbb{P}, \boldsymbol{r}, \mathcal{W})$. $\mathcal{S}$ is a finite *state* set where $|\mathcal{S}| = S$. $\mathcal{A}$ is a finite *action* set where $|\mathcal{A}| = A$. $H$ is the length of the horizon. $\mathbb{P}(\cdot \mid x, a)$ is a *stationary, unknown transition probability* to a new state for taking action $a$ at state $x$. $\boldsymbol{r} = \{\boldsymbol{r}_1, \ldots, \boldsymbol{r}_H\}$, where $\boldsymbol{r}_h : \mathcal{S} \times \mathcal{A} \to [0, 1]^d$ represents a $d$-dimensional *vector rewards function* that captures the $d$ objectives[2]. $\mathcal{W} := \{w \in [0, 1]^d, \|w\|_1 = 1\}$ specifies the set of all possible *preference vectors*[3], where each preference vector $w \in \mathcal{W}$ induces a scalar reward function by[4] $r_h(\cdot, \cdot) = \langle w, \boldsymbol{r}_h(\cdot, \cdot) \rangle$ for $h = 1, \ldots, H$. A *policy* is represented by $\pi := \{\pi_1, \ldots, \pi_H\}$, where each $\pi_h(\cdot)$ maps a state to a distribution over the action set. Fixing a policy $\pi$, we will consider the following generalized *Q-value function* and generalized *value function*:

$$Q_h^\pi(x, a; w) := \mathbb{E}\left[\sum_{j=h}^H \langle w, \boldsymbol{r}_j(x_j, a_j) \rangle \big| x_h = x, a_h = a\right], \quad V_h^\pi(x; w) := Q_h^\pi(x, \pi_h(x); w),$$

where $x_j \sim \mathbb{P}(\cdot \mid x_{j-1}, a_{j-1})$ and $a_j \sim \pi_j(x_j)$ for $j > h$. Note that compared with the typical $Q$-value function (or the value function) used in single-objective RL, here the generalized $Q$-value function (or the generalized value function) takes the preference vector as an additional input, besides the state-action pair. Fixing a preference $w \in \mathcal{W}$, the optimal policy under $w$ is defined as $\pi_w^* := \arg\max_\pi V_1^\pi(x_1; w)$. For simplicity, we denote

$$Q_h^*(x, a; w) := Q_h^{\pi_w^*}(x, a; w), \quad V_h^*(x; w) := V_h^{\pi_w^*}(x; w) = \max_a Q_h^*(x, a; w).$$

The following abbreviation is adopted for simplicity:

$$\mathbb{P}V_h^\pi\big|_{x,a,w} := \sum_{y \in \mathcal{S}} \mathbb{P}(y|x, a) V_h^\pi(y; w),$$

and similar abbreviations will be adopted for variants of probability transition (e.g., the empirical transition probability) and variants of value function (e.g., the estimated value function). Finally, we remark that the following Bellman equations hold for the generalized $Q$-value functions

$$\begin{cases} Q_h^\pi(x, a; w) = \langle w, \boldsymbol{r}_h(x, a) \rangle + \mathbb{P}V_{h+1}^\pi\big|_{x,a,w}, \\ V_h^\pi(x; w) = Q_h^\pi(x, \pi_h(x); w); \end{cases} \quad \begin{cases} Q_h^*(x, a; w) = \langle w, \boldsymbol{r}_h(x, a) \rangle + \mathbb{P}V_{h+1}^*\big|_{x,a,w}, \\ V_h^*(x; w) = \max_a Q_h^*(x, a; w). \end{cases}$$

With the above preparations, we are ready to discuss our algorithms and theory for online MORL (Section 3) and preference-free exploration (Section 4).

## 3 Online MORL

**Problem Setups.** The online setting captures the first difficulty in MORL, where the preference vectors can be adversarially provided to the MORL agent. Formally, the MORL agent interacts with

---

[2]For the sake of presentation, we discuss bounded deterministic reward functions in this work. The techniques can be readily extended to stochastic reward settings.

[3]The condition that $w \in [0, 1]^d$ is only assumed for convenience. Our results naturally generalize to preference vectors that are entry-wisely bounded by absolute constants.

[4]The linear scalarization method can be generalized. See more discussions in Section 3.2, Remark 2.

an unknown environment through Protocol 1: at the beginning of the $k$-th episode, an adversary selects a preference vector $w^k$ and reveals it to the agent; then starting from a fixed initial state[5] $x_1$, the agent draws a trajectory from the environment by recursively taking an action and observing a new state, and collects rewards from the trajectory, where the rewards are scalarized from the vector rewards by the given preference. The agent's goal is to find the policies that maximize the cumulative rewards. Its performance will be measured by the following *regret*: suppose the MORL agent has interacted with the environment through Protocol 1 for $K$ episodes, where the provided preferences are $\{w^k\}_{k=1}^K$ and the adopted policies are $\{\pi^k\}_{k=1}^K$ correspondingly, we consider the regret of the collected rewards (in expectation) competing with the theoretically maximum collected rewards (in expectation):

$$\texttt{regret}(K) := \sum_{k=1}^K V_1^*(x_1; w^k) - V_1^{\pi^k}(x_1; w^k). \tag{1}$$

Clearly, the regret (1) is always non-negative, and a smaller regret implies a better online performance. We would like to highlight that the regret (1) allows the theoretically optimal agent to adopt varying policies that adapt to the preferences.

---

**Protocol 1** Environment Interaction Protocol

**Require:** MDP$(\mathcal{S}, \mathcal{A}, H, \mathbb{P}, \boldsymbol{r}, \mathcal{W})$ and online agent
 1: agent observes $(\mathcal{S}, \mathcal{A}, H, \boldsymbol{r}, \mathcal{W})$
 2: **for** episode $k = 1, 2, \ldots, K$ **do**
 3:     agent receives an initial state $x_1^k = x_1$, and a preference $w^k$ (from an adversary)
 4:     **for** step $h = 1, 2, \ldots, H$ **do**
 5:         agent chooses an action $a_h^k$, and collects reward $\langle w^k, \boldsymbol{r}_h(x_h^k, a_h^k) \rangle$
 6:         agent transits to a new state $x_{h+1}^k \sim \mathbb{P}(\cdot \mid x_h^k, a_h^k)$
 7:     **end for**
 8: **end for**

---

**Connections to Online Single-Objective RL.** The online regret minimization problems are well investigated in the context of single-objective RL (see, e.g., [Azar et al., 2017, Jin et al., 2018]). In both online single-objective RL and our studied online MORL, it is typical to assume that the transition probability $\mathbb{P}$ is the only unknown information about the environment since estimating a stochastic reward is relatively easy (see, e.g., [Azar et al., 2017, Jin et al., 2018]). In particular, single-objective RL is a special case of MORL in the online setting, where the preference is fixed during the entire learning process (i.e., $w^k := w$ for all $k$) — therefore an online MORL algorithm naturally applies to single-objective RL. However, the reverse is not true as that in MORL the preference vectors can change over time and are potentially adversarial.

**Comparison with Single-Objective Stochastic and Adversarial Reward Setting.** The essential difficulty of MORL is further reflected in the regret (1). Specifically, the regret (1) compares the perfor-

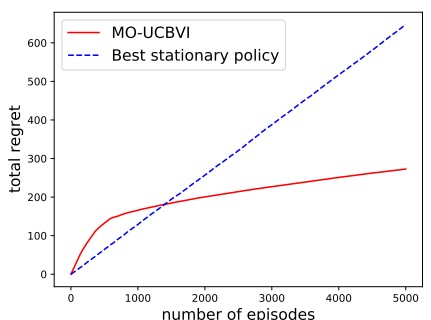

Figure 1: A regret comparison of `MO-UCBVI` vs. the best-in-hindsight policy in a simulated random multi-objective MDP. Note that the best-in-hindsight policy is the optimal policy for single-objective RL. The plots show that the best-in-hindsight policy will incur linear regret in online MORL, and the proposed `MO-UCBVI` achieves sublinear regret as predicted by Theorem 1. See Appendix A for details.

mance of the agent's policies to a sequence of *optimal* policy under *each* given preference, which could *vary* over time. However, in online single-objective RL, either with known/stochastic rewards [Azar et al., 2017, Jin et al., 2018] or adversarial rewards [Rosenberg and Mansour, 2019, Jin et al., 2019], the benchmark policy is supposed to be *fixed* over time (the best policy in the hindsight). This difference suggests that online MORL could be more challenging than online single-objective RL, as a harder performance measurement is adopted. More specifically, when measured by regret

---

[5]Without loss of generality, we fix the initial state; otherwise we may as well consider an MDP with an external initial state $x_0$ with zero reward for all actions, and a transition $\mathbb{P}_0(\cdot \mid x_0, a) = \mathbb{P}_0(\cdot)$ for all action $a$. This is equivalent to our setting by letting the horizon length $H$ be $H + 1$.

(1), existing algorithms for single-objective RL (e.g., [Azar et al., 2017]) easily suffer a $\propto \Theta(K)$ regret when the sequence of preferences is adversarially designed; in contrast, we will show an MORL algorithm that experiences at most $\propto \widetilde{\mathcal{O}}(\sqrt{K})$ regret under any sequence of preferences. A numerical simulation of this issue is presented in Figure 1.

### 3.1 A Sample-Efficient Online Algorithm

In this part we introduce an algorithm for online MORL, called *multi-objective upper confidence bound value iteration* (`MO-UCBVI`). A simplified verison of `MO-UCBVI` is presented as Algorithm 1, and a more advanced version (Algorithm 4) can be found in Appendix B. Our algorithm is inspired by `UCBVI` [Azar et al., 2017] that achieves minimax regret bound in online single-objective RL.

---

**Algorithm 1** `MO-UCBVI`

---

1: initialize history $\mathcal{H}^0 = \emptyset$
2: **for** episode $k = 1, 2, \ldots, K$ **do**
3:      $N^k(x, a), \widehat{\mathbb{P}}^k(y \mid x, a) \leftarrow$ `Empi-Prob`$(\mathcal{H}^{k-1})$
4:      compute bonus $b^k(x, a) := c \cdot \sqrt{\frac{\min\{d, S\} H^2 \iota}{N^k(x,a)}}$, where $\iota = \log(HSAK/\delta)$ and $c$ is a constant
5:      receive a preference $w^k$
6:      $\{Q_h^k(x, a; w^k)\}_{h=1}^H \leftarrow$ `UCB-Q-Value`$(\widehat{\mathbb{P}}^k, w^k, b^k)$
7:      receive initial state $x_1^k = x_1$
8:      **for** step $h = 1, 2, \ldots, H$ **do**
9:          take action $a_h^k = \arg\max_a Q_h^k(x_h^k, a; w^k)$, and obtain a new state $x_{h+1}^k$
10:      **end for**
11:      update history $\mathcal{H}^k = \mathcal{H}^{k-1} \cup \{x_h^k, a_h^k\}_{h=1}^H$
12: **end for**

13: **Function** `Empi-Prob`
14: **Require:** history $\mathcal{H}^{k-1}$
15: **for** $(x, a, y) \in \mathcal{S} \times \mathcal{A} \times \mathcal{S}$ **do**
16:      $N^k(x, a, y) := \#\{(x, a, y) \in \mathcal{H}^{k-1}\}$, and $N^k(x, a) := \sum_y N^k(x, a, y)$
17:      **if** $N^k(x, a) > 0$ **then**
18:          $\widehat{\mathbb{P}}^k(y \mid x, a) = N^k(x, a, y)/N^k(x, a)$
19:      **else**
20:          $\widehat{\mathbb{P}}^k(y \mid x, a) = 1/S$
21:      **end if**
22: **end for**
23: **return** $N^k(x, a)$ and $\widehat{\mathbb{P}}^k(y \mid x, a)$

24: **Function** `UCB-Q-Value`
25: **Require:** empirical transition $\widehat{\mathbb{P}}^k$, preference $w^k$, and bonus $b^k(x, a)$
26: set $V_{H+1}^k(x; w^k) = 0$
27: **for** step $h = H, H-1, \ldots, 1$ **do**
28:      **for** $(x, a) \in \mathcal{S} \times \mathcal{A}$ **do**
29:          $Q_h^k(x, a; w^k) = \min\left\{H, \langle w^k, \boldsymbol{r}_h(x, a)\rangle + b^k(x, a) + \widehat{\mathbb{P}}_h^k V_{h+1}^k\big|_{x,a,w^k}\right\}$
30:          $V_h^k(x; w^k) = \max_{a \in \mathcal{A}} Q_h^k(x, a; w^k)$
31:      **end for**
32: **end for**
33: **return** $\left\{Q_h^k(x, a; w^k)\right\}_{h=1}^H$

---

In Algorithm 1, the agent interacts with the environment according to Protocol 1, and use an *optimistic policy* to explore the unknown environment and collect rewards. The optimistic policy is a greedy policy with respect to an optimistic estimation to the value function, `UCB-Q-Value` (lines 6 and 9). Specifically, `UCB-Q-Value` is constructed to maximize the cumulative reward, which is scalarized by the current preference, through dynamic programming over an empirical transition probability (line 24). The empirical transition probability is inferred from the data collected so far (lines 3 and

13), which might not be accurate if a state-action pair has not yet been visited for sufficient times. To mitigate this inaccuracy, `UCB-Q-Value` utilizes an extra *exploration bonus* (lines 4 and 29) so that: (i) `UCB-Q-Value` never under-estimates the optimal true value function, for *whatever preference vector* (and with high probability); and (ii) the added bonus *shrinks quickly* enough (as the corresponding state-action pair continuously being visited) so that the over-estimation is under control. The overall consequence is that: `MO-UCBVI` explores the environment via a sequence of optimistic policies, in order to collect rewards under a sequence of adversarially provided preferences; since the policies are optimistic for any preference, the incurred regret would not exceed the total amount of added bonus; and since the bonus decays fast, their sum up would be sublinear (with respect to the number of episodes played). Therefore `MO-UCBVI` only suffers a sublinear regret, even when the preferences are adversarially presented. The above intuition is formalized in the next part.

## 3.2 Theoretical Analysis

The next two theorems justify regret upper bounds for `MO-UCBVI` and a regret lower bound for online MORL problems, respectively.

**Theorem 1** (Regret bounds for `MO-UCBVI`). *Suppose $K$ is sufficiently large. Then for any sequence of the incoming preferences $\{w^k\}_{k=1}^K$, with probability at least $1 - \delta$:*

- *the regret (1) of `MO-UCBVI` (see Algorithm 1) satisfies*

$$\texttt{regret}(K) \leq \mathcal{O}\big(\sqrt{\min\{d, S\} \cdot H^3 SAK \log(HSAK/\delta)}\big);$$

- *and the regret (1) of a Bernstein-variant `MO-UCBVI` (see Algorithm 4 in Appendix B) satisfies*

$$\texttt{regret}(K) \leq \mathcal{O}\big(\sqrt{\min\{d, S\} \cdot H^2 SAK \log^2(HSAK/\delta)}\big).$$

**Theorem 2** (A regret lower bound for MORL). *There exist some absolute constants $c, K_0 > 0$, such that for any $K > K_0$, any MORL algorithm that runs $K$ episodes, there is a set of MOMDPs and a sequence of (necessarily adversarially chosen) preferences vectors such that*

$$\mathbb{E}[\texttt{regret}(K)] \geq c \cdot \sqrt{\min\{d, S\} \cdot H^2 SAK},$$

*where the expectation is taken with respect to the randomness of choosing MOMDPs and the randomness of the algorithm for collecting dataset.*

*Remark* 1. When $d = 1$, MORL recovers the single-objective RL setting, and Theorem 1 recovers existing nearly minimax regret bounds for single-objective RL [Azar et al., 2017, Zanette and Brunskill, 2019]. Moreover, the lower bound in Theorem 2 implies that our upper bound in Theorem 1 is tight ignoring logarithm terms. Interestingly, the lower bound suggests MORL with $d > 2$ is truly harder than single objective RL (corresponding to $d = 1$) as the sequence of preferences can be adversarially chosen.

*Remark* 2. Theorem 1 (as well as our other theorems) applies to general scalarization methods besides the linear one as adopted by Algorithm 1 and many other MORL papers [Yang et al., 2019, Roijers et al., 2013, Natarajan and Tadepalli, 2005, Abels et al., 2018]. In particular, our results apply to scalarization functions $r_h(\cdot, \cdot) = f(\boldsymbol{r}_h(\cdot, \cdot); w)$ that are (1) deterministic, (2) Lipschitz continuous for $w$, and (3) bounded between $[0, 1]$ (which can be relaxed). This will be clear from proofs in Appendix B, where we treat the potentially adversarially given preferences by a covering argument and union bound, and these techniques are not dedicated to linear scalarization function and can be easily extended to more general cases.

The proof of Theorem 1 leverages standard analysis procedures for single-objective RL [Azar et al., 2017, Zanette and Brunskill, 2019], and a covering argument with an union bound to tackle the challenge of adversarial preferences. Th rigorous proof is included in Appendix B.

Specifying an adversarial process of providing preferences is the key challenge for proving Theorem 2. To handle this issue, we use reduction techniques and utilize a lower bound that we will show shortly for preference-free exploration problems. We refer the readers to Theorem 4 and Appendix E for more details.

## 4 Preference-Free Exploration

**Problem Setups.** *Preference-free exploration* (PFE) captures the second difficulty in MORL: the preference vector might not be observable when the agent explores the environment. Specifically, PFE

consists of an exploration phase and a planning phase. Similarly as in the online setting, the transition probability is hidden from the MORL agent. In the exploration phase, the agent interacts with the unknown environment to collect samples, however the agent has no information about the preference vectors at this point. Afterwards PFE switches to the planning phase, where the agent is prohibited to obtain new data, and is required to compute near-optimal policy for any preference-weighted reward functions. Since this task is no longer in an online fashion, we turn to measure the performance of a PFE algorithm by the minimum number of required trajectories (in the exploration phase) so that the algorithm can behave near-optimally in the planning phase. This is made formal as follows: a PFE algorithm is called $(\epsilon, \delta)$-PAC (Probably Approximately Correct), if

$$\mathbb{P}\big\{\forall w \in \mathcal{W}, \ V_1^*(x_1; w) - V_1^{\pi_w}(x_1; w) \le \epsilon\big\} \ge 1 - \delta,$$

where $\pi_w$ is the policy outputted by the PFE algorithm for input preference $w$, then the *sample complexity* of a PFE algorithm is defined by the least amount of trajectories it needs to collect in the exploration phase for being $(\epsilon, \delta)$-PAC in the planning phase.

**Connections to Reward-Free Exploration.** PFE problem is a natural extension to the recent proposed *reward-free exploration* (RFE) problem [Jin et al., 2020, Kaufmann et al., 2020, Wang et al., 2020, Ménard et al., 2020, Zhang et al., 2020b]. Both problems consist of an exploration phase and a planning phase; the difference is in the planning phase: in RFE, the agent needs to be able to compute near-optimal policies for *all reward functions*, while in PFE, the agent only needs to achieve that for *all preference-weighted reward functions*, i.e., the reward functions that can be represented as the inner product of a $d$-dimensional preference vectors and the $d$-dimensional vector rewards functions (i.e., the $d$ objectives in MORL). A PFE problem reduces to a RFE problem if $d = SA$ such that every reward function can be represented as a preference-weighted reward function. However, if $d \ll SA$, it is conjectured by Jin et al. [2020] that PFE can be solved with a much smaller sample complexity than RFE. Indeed, in the following part we show an algorithm that improves a $\propto \widetilde{\mathcal{O}}(S^2)$ dependence for RFE to $\propto \widetilde{\mathcal{O}}(\min\{d, S\} \cdot S)$ dependence for PFE, in terms of sample complexity.

### 4.1 A Sample-Efficient Exploration Algorithm

We now present a simple variant of `MO-UCBVI` that is sample-efficient for PFE. The algorithm is called *preference-free upper confidence bound exploration* (`PF-UCB`), and is discussed separately as in the exploration phase (Algorithm 2) and in the planning phase (Algorithm 3) in below.

---

**Algorithm 2** `PF-UCB` (Exploration)

---

1: initialize history $\mathcal{H}^0 = \emptyset$
2: **for** episode $k = 1, 2, \ldots, K$ **do**
3:      $N^k(x, a), \ \widehat{\mathbb{P}}^k(y \mid x, a) \leftarrow \texttt{Empi-Prob}(\mathcal{H}^{k-1})$
4:      compute bonus $c^k(x, a) := \frac{H^2 S}{2N^k(x,a)} + 2b^k(x, a)$ for $b^k(x, a)$ defined in Algorithms 1 or 3
5:      $\{\overline{Q}_h^k(x, a)\}_{h=1}^H \leftarrow \texttt{UCB-Q-Value}(\widehat{\mathbb{P}}^k, w = 0, c^k)$
6:      receive initial state $x_1^k = x_1$
7:      **for** step $h = 1, 2, \ldots, H$ **do**
8:          take action $a_h^k = \arg\max_a \overline{Q}_h^k(x_h^k, a)$, and obtain a new state $x_{h+1}^k$
9:      **end for**
10:     update history $\mathcal{H}^k = \mathcal{H}^{k-1} \cup \{x_h^k, a_h^k\}_{h=1}^H$
11: **end for**

---

Algorithm 2 presents our PFE algorithm in the exploration phase. Indeed, Algorithm 2 is a modified `MO-UCBVI` (Algorithm 1) by setting the preference to be zero (Algorithm 2, line 5), and slightly enlarging the exploration bonus (Algorithm 2, line 4). The intention of an increased exploration bonus will be made clear later when we discuss the planning phase. With a zero preference vector, the `UCB-Q-Value` in Algorithm 2 will identify a trajectory along which the cumulative bonus (instead of the cumulative rewards) is maximized (with respect to the empirical transition probability). Also note that the bonus function (Algorithm 2, line 4) is negatively correlated with the number of visits to a state-action pair. Hence the greedy policy with respect to the zero-preference `UCB-Q-Value` tends to visit the state-actions pairs that are associated with large bonus, i.e., those that have been visited

---

**Algorithm 3** `PF-UCB` (Planning)

---

**Require:** history $\mathcal{H}^K$, preference vector $w$

1: **for** $k = 1, 2, \ldots, K$ **do**
2:    $N^k(x, a), \widehat{\mathbb{P}}^k(y|x, a) \leftarrow$ `Empi-Prob`$(\mathcal{H}^{k-1})$
3:    compute bonus $b^k(x, a) := c \cdot \sqrt{\frac{\min\{d, S\} H^2 \iota}{N^k(x, a)}}$ where $\iota = \log(HSAK/\delta)$ and $c$ is a constant
4:    $\{Q_h^k(\cdot, \cdot; w)\}_{h=1}^H \leftarrow$ `UCB-Q-Value`$(\widehat{\mathbb{P}}^k, w, b^k)$
5:    infer greedy policy $\pi_h^k(x) = \arg\max_a Q_h^k(x, a; w)$
6: **end for**
7: **return** $\pi$ drawn uniformly from $\{\pi^k\}_{k=1}^K$

---

for less times. In sum, Algorithm 2 explores the unknown environment "uniformly", without the guidance of preference vectors.

Then Algorithm 3 shows our PFE algorithm in the planning phase. Given any preference vector, Algorithm 3 computes a sequence of optimistically estimated value functions based on the data collected from the exploration phase, and then outputs a greedy policy with respect to a randomly drawn optimistic value estimation. Note that the bonus in Algorithm 3 is set as the one in Algorithm 1, and recall that the bonus in Algorithm 2 is an enlarged one. The relatively large bonus in the exploration phase guarantees that the regret in the planning phase never exceeds that in the exploration phase. On the other hand, based on Theorem 1 for `MO-UCBVI` (Algorithm 1), we have already known that the exploration algorithm (Algorithm 2), a modified Algorithm 1, suffers at most $\widetilde{\mathcal{O}}(\sqrt{K})$ regret, hence the planning algorithm (Algorithm 3) experiences at most $\widetilde{\mathcal{O}}(1/\sqrt{K})$ error. The next part rigorously justifies these discussions.

### 4.2 Theoretic Analysis

We first provide Theorem 3 to justify the trajectory complexity of Algorithms 2 and 3 in the PFE setting; then we present Theorem 4 that gives an information-theoretic lower bound on the trajectory complexity for any PFE algorithm.

**Theorem 3** (A trajectory complexity of `PF-UCB`). *Suppose $\epsilon > 0$ is sufficiently small. Then for `PF-UCB` (Algorithm 2) run for*

$$K = \mathcal{O}\big(\min\{d, S\} \cdot H^3 SA\iota/\epsilon^2\big), \quad \text{where } \iota := \log(HSA/(\delta\epsilon)),$$

*episodes, `PF-UCB` (Algorithm 3) is $(\epsilon, \delta)$-PAC.*

**Theorem 4** (A lower bound for PFE). *There exist absolute constants $c, \epsilon_0 > 0$, such that for any $0 < \epsilon < \epsilon_0$, there exists a set of MOMDPs such that any PFE algorithm that is $(\epsilon, 0.1)$-PAC on them, it needs to collect at least*

$$K \geq \Omega\big(\min\{d, S\} \cdot H^2 SA/\epsilon^2\big)$$

*trajectories in expectation (with respect to the randomness of choosing MOMDPs and the exploration algorithm).*

*Remark* 3. According to Theorems 3 and 4, the trajectory complexity of `PF-UCB` is optimal for $d, S, A, \epsilon$ ignoring logarithmic factors, but is an $H$ factor loose compared with the lower bound. This is because the current Algorithm 2 utilizes a *preference-independent*, Hoeffding-type bonus since the preference vector is not available during exploration. We leave it as an open problem to further remove this gap about $H$.

**Proof Sketch of Theorem 3.** Theorem 3 is obtained in three procedures. (1) We first observe the total regret incurred by Algorithm 2 is $\widetilde{\mathcal{O}}\big(\sqrt{\min\{d, S\} H^3 SAK}\big)$ according to Theorem 1. (2) Then utilizing the enlarged exploration bonus, we show that in each episode, the planning error is at most constant times of the incurred exploration error. (3) Thus the averaged planning error is at most $\widetilde{\mathcal{O}}\big(\sqrt{\min\{d, S\} H^3 SA/K}\big)$ as claimed. A complete proof is deferred to Appendix C.

Note that the second argument is motivated by [Zhang et al., 2020a, Wang et al., 2020]. However in their original paper, a brute-force union bound over all possible value functions are required to obtain similar effect, due to the limitation of model-free algorithm [Zhang et al., 2020a] (see Appendix F

for more details) or linear function approximation [Wang et al., 2020]. This will cause a loose, $\propto \widetilde{\mathcal{O}}(S^2)$ complexity in the obtained bound. Different from their approach, we carefully manipulate a lower order term to avoid union bounding all value functions during the second argument. As a consequence we obtain a near-tight $\propto \widetilde{\mathcal{O}}(\min\{d, S\} \cdot S)$ dependence in the final bound. We believe this observation has broader application in the analysis of similar RL problems.

**Proof Sketch of Theorem 4.** We next introduce the idea of constructing the hard instance that witnesses the lower bound in Theorem 4. A basic ingredient is the hard instance given by Jin et al. [2020]. However, this hard instance is invented for RFE, where the corresponding lower bound is $K \geq \Omega\big(H^2 S^2 A / \epsilon^2\big)$. Note this lower bound cannot match the upper bound in Theorem 3 in terms of $d$ and $S$. In order to develop a dedicated lower bound for PFE, we utilize Johnson–Lindenstrauss Lemma [Johnson and Lindenstrauss, 1984] to refine the hard instance in [Jin et al., 2020], and successfully reduce a factor $S$ to $\min\{d, S\}$ in their lower bound, which gives the result in Theorem 4. We believe that the idea to refine RL hard instance by Johnson–Lindenstrauss Lemma is of broader interests. A rigorous proof is deferred to Appendix D.

**Application in Reward-Free Exploration.** By setting $d = SA$ and allowing arbitrary reward functions, PFE problems reduce to RFE problems. Therefore as a side product, Theorem 3 implies the following results for RFE problems on stationary or non-stationary MDPs[6]:

**Corollary 5.** *Suppose $\epsilon > 0$ is sufficiently small. Consider the reward-free exploration problems on a* stationary *MDP. Suppose* `PF-UCB` *(Algorithm 2) is run for*

$$K = \mathcal{O}\big(H^3 S^2 A \iota / \epsilon^2\big), \quad \text{where } \iota := \log(HSA/(\delta\epsilon)),$$

*episodes, then* `PF-UCB` *(Algorithm 3) is $(\epsilon, \delta)$-PAC. Moreover, if the MDP is* non-stationary, *the above bound will be revised to $K = \mathcal{O}\big(H^4 S^2 A \iota / \epsilon^2\big)$.*

*Remark* 4. When applied to RFE, `PF-UCB` matches the rate shown in [Kaufmann et al., 2020] and improves an $H$ factor compared with [Jin et al., 2020] (for both stationary and non-stationary MDPs), but is an $H$ factor loose compared with the current best rates, [Ménard et al., 2020] (for non-stationary MDPs) and [Zhang et al., 2020b] (for stationary MDPs). However we highlight that our results are superior in the context of PFE, since `PF-UCB` adapts with the structure of the rewards. In specific, if rewards admit a $d$-dimensional feature for $d < S$, `PF-UCB` only needs $\propto \widetilde{\mathcal{O}}(\min\{d, S\} \cdot S)$ samples, but the above methods must explore the whole environment with a high precision which consumes $\propto \widetilde{\mathcal{O}}(S^2)$ samples.

**Application in Task-Agnostic Exploration.** PFE is also related to *task-agnostic exploration* (TAE) [Zhang et al., 2020a]: in PFE, the agent needs to plan for an arbitrary reward function from a fixed and bounded $d$-dimensional space; and in TAE, the agent needs to plan for $N$ fixed reward functions. Due to the nature of the problem setups, PFE algorithms (that do not exploit the given reward basis $\boldsymbol{r}$ during exploration, e.g., ours) and TAE algorithms can be easily applied to solve the other problem through a covering argument and a union bound, and with a modification of $\min\{d, S\} \leftrightarrow \log(N)$ in the obtained trajectory complexity bounds. For TAE on a non-stationary MDP, Zhang et al. [2020a] show an algorithm which takes $\widetilde{\mathcal{O}}\big(\log(N) \cdot H^5 S A / \epsilon^2\big)$ episodes for TAE[7]. In comparison, when applied to TAE on a non-stationary MDP, Theorem 3 implies `PF-UCB` only takes $\widetilde{\mathcal{O}}\big(\log(N) \cdot H^4 S A / \epsilon^2\big)$ episodes[8], which improves [Zhang et al., 2020a].

---

[6]Our considered MDP is *stationary* as the transition probability $\mathbb{P}$ is fixed (across different steps). An MDP is called *non-stationary*, if the transition probability varies at different steps, i.e., replacing $\mathbb{P}$ by $\{\mathbb{P}_h\}_{h=1}^{H}$.

[7]This bound is copied from [Zhang et al., 2020a], which is erroneously stated due to a technical issue in the proof of Lemma 2 in their original paper. The issue can be fixed by a covering argument and union bound on the value functions, but then the obtained bound should be $\widetilde{\mathcal{O}}\big(\log(N) H^5 S^2 A / \epsilon^2\big)$. See Appendix F for details.

[8]The conversion holds as follows. First set $d = 1$ in our algorithm to yield an algorithm for TAE with a single agnostic task, where we have $\min\{d, S\} = 1$. Then one can extend this algorithm to TAE with $N$ agnostic tasks using a union bound to have the algorithm succeed simultaneously for all $N$ tasks, which adds a $\log N$ multiplicative factor in the sample complexity bound. In this way, we obtain a TAE algorithm with a sample complexity bound in Theorem 3 where $\min\{d, S\}$ is replaced with $\log N$.

# 5 Related Works

**MORL.** MORL receives extensive attention from RL applications [Yang et al., 2019, Natarajan and Tadepalli, 2005, Mossalam et al., 2016, Abels et al., 2018, Roijers et al., 2013]. However, little theoretical results are known, especially in terms of the sample efficiency and adversarial preference settings. A large amount of MORL works focus on identifying (a cover for) the policies belonging to the *Pareto front* [Yang et al., 2019, Roijers et al., 2013, Cheung, 2019], but the correspondence between a preference and an optimal policy is ignored. Hence they cannot "accommodate picky customers" as our algorithms do. To our knowledge, this paper initiates the theoretical study of the sample efficiency for MORL in the setting of adversarial preferences.

**Adversarial MDP.** Similarly to the online MORL problem studied in this paper, the adversarial MDP problem [Neu et al., 2012, Rosenberg and Mansour, 2019, Jin et al., 2019] also allows the reward function to change adversarially over time. However, we study a totally different regret (see (1)). In the adversarial MDP problem, the regret is measured against a *fixed* policy that is best in the hindsight; but in online MORL problem, we study a regret against a sequence of optimal policies with respect to the sequence of incoming preferences, i.e., our benchmark policy may *vary* over time. Therefore our regret bound for online MORL problems is orthogonal to those for adversarial MDP problems [Neu et al., 2012, Rosenberg and Mansour, 2019, Jin et al., 2019].

**Constrained MDP.** In the problem of the *constrained MDP*, $d$ constraints are enforced to restrict the policy domain, and the agent aims to find a policy that belongs to the domain and maximizes the cumulative rewards [Achiam et al., 2017, Efroni et al., 2020, El Chamie et al., 2018, Fisac et al., 2018, Wachi and Sui, 2020, García and Fernández, 2015, Brantley et al., 2020]. Constrained MDP is related to MORL as when the constraints are soft, they can be formulated as "objectives" with negative weights. However, there is a fundamental difference: constrained MDP aims to optimize only one (and known) objective, where in MORL studied in this paper, we aim to be able to find near optimal policies for any preference-weighted objective (to accommodate picky customers).

**Reward-Free Exploration.** The proposed preference-free exploration problem generalizes the problems of reward-free exploration [Jin et al., 2020]. Compared with existing works for RFE [Jin et al., 2020, Kaufmann et al., 2020, Ménard et al., 2020, Zhang et al., 2020b], our method has the advantage of adapting with rewards that admit low-dimensional structure — this partly answers an open question raised by Jin et al. [2020]. We also note that Wang et al. [2020] study RFE with linear function approximation on the value functions; in contrast our setting can be interpreted as RFE with linear function approximation on the reward functions, which is orthogonal to their setting.

# 6 Conclusion

In this paper we study provably sample-efficient algorithms for multi-objective reinforcement learning in both online and preference-free exploration settings. For both settings, sample-efficient algorithms are proposed and their sample complexity analysis is provided; moreover, two information-theoretic lower bounds are proved to justify the near-tightness of the proposed algorithms, respectively. Our results extend existing theory for single-objective RL and reward-free exploration, and resolve an open question raised by Jin et al. [2020].

## Acknowledgement

This research was supported in part by NSF CAREER grant 1652257, ONR Award N00014-18-1-2364, the Lifelong Learning Machines program from DARPA/MTO, and NSF HDR TRIPODS grant 1934979.

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
