Figure 2: `MO-UCBVI` vs. Q-learning [Jin et al., 2018] in a simulated random multi-objective MDP. The plot suggests that `MO-UCBVI` achieves sublinear regret but Q-learning incurs linear regret.

## A  Numerical Simulations

Code for simulations is available at `https://github.com/uuujf/MORL`.

**Comparison with the Optimal Single-Objective RL Algorithm.**  Figure 1 shows a regret comparison between the proposed `MO-UCBVI` and the best-in-hindsight policy. In the experiment we simulate a random multi-objective MDP with $S = 20, A = 5, H = 10, d = 15$, and run the algorithms for $K = 5000$ episodes. The best-in-hindsight policy refers to a policy that is fixed across episodes and achieves maximum cumulative rewards in the whole game, i.e., $\arg\max_\pi \sum_{k=1}^K V_1^k(x_1; w^k)$. Note that the best-in-hindsight policy is the optimal policy in the single-objective RL setting; however it could be much worse than a time-varying policy in the multi-objective RL setting. Since the information-theoretically adversarial preferences are computationally infeasible to compute, in Figure 1 we simply use a randomly generated set of preferences, for which the "best-in-hindsight" policy already performs poorly. In sum, the plot shows that our algorithm achieves sublinear regret in online MORL, but the best single-objective RL algorithm will incur linear regret in online MORL.

**Comparison with Q-Learning.**  Since MORL is connected to task-agnostic exploration [Zhang et al., 2020a] in the exploration setting (see discussions in Section 4), it is tempted to think that the Q-learning method studied in Jin et al. [2018], Zhang et al. [2020a] could also work in the setting of online MORL. However this is refuted by Figure 2. We simulate a random multi-objective MDP with $S = 20, A = 5, H = 10, d = 15$, and run the algorithms for $K = 5000$ episodes. The Q-learning algorithm is specified by Algorithm 1 in Jin et al. [2018], excepted that the reward is now a linear scalarization of a preference vector and a multi-objective reward vector. Figure 2 shows that Q-learning cannot achieve a sublinear regret in the setting of online MORL.

**The Effect of Number of Objectives.**  Figure 3 shows the performance of `MO-UCBVI` with different number of objectives. In the experiment we simulate a random multi-objective MDP with $S = 20, A = 5, H = 10$, and number of objectives $d \in \{1, 5, 15, 20, 30\}$, and run `MO-UCBVI` for $K = 5000$ episodes. As shown in Figure 3, the regret will increase as the number of objectives increases; moreover, in all settings `MO-UCBVI` achieves a sublinear regret.

## B  Proof of Theorem 1

Our proof utilizes techniques from [Zanette and Brunskill, 2019].

**Notations.**  For two functions $f(x) \geq 0$ and $g(x) \geq 0$ defined for $x \in [0, \infty)$, we write $f(x) \lesssim g(x)$ if $f(x) \leq c \cdot g(x)$ for some absolute constant $c > 0$; we write $f(x) \gtrsim g(x)$ if $g(x) \lesssim f(x)$; and we write $f(x) \asymp g(x)$ if $f(x) \lesssim g(x) \lesssim f(x)$. Moreover, we write $f(x) = \mathcal{O}(g(x))$ if $\lim_{x\to\infty} f(x)/g(x) < c$ for some absolute constant $c > 0$; we write $f(x) = \Omega(g(x))$ if $g(x) = \mathcal{O}(f(x))$; and we write $f(x) = \Theta(g(x))$ if $f(x) = \mathcal{O}(g(x))$ and $g(x) = \mathcal{O}(f(x))$. To hide

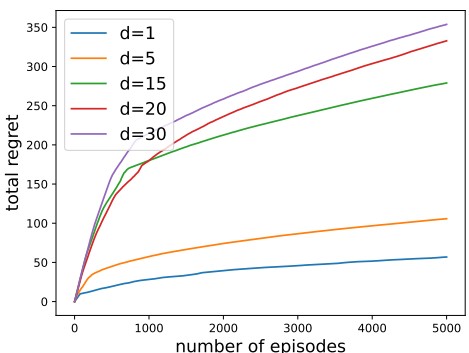

Figure 3: The effect of number of objectives. The plot shows the performance of `MO-UCBVI` in a simulated random multi-objective MDP with different number of objectives. The plot suggests that regret increases as number of objectives increases, but `MO-UCBVI` achieves sublinear regret in all cases, as predicted by Theorem 1.

the logarithmic factors, we write $f(x) = \widetilde{\mathcal{O}}(g(x))$ if $f(x) = \mathcal{O}(g(x) \log^d x)$ for some absolute constant $d > 0$. For $a, b \in \mathbb{R}$, we write $a \wedge b := \min\{a, b\}$ and $a \vee b := \max\{a, b\}$. We will use $\iota$ to denote a general logarithmic factor. Let $\pi^k$ be the planning policy at the $k$-th episode, i.e., a greedy policy that maximizes $Q_h^k(x, a)$. Let $w_h^k(x, a) := \mathbb{P}\left\{(x_h, a_h) = (x, a) \mid \pi^k, \mathbb{P}\right\}$ and $w^k(x, a) := \sum_h w_h^k(x, a)$.

## B.1  Good Events and Useful Lemmas

Fix $\epsilon$ to be a constant to be determined later (we will set $\epsilon = 1/K$). Consider the following events:

$$G_H := \left\{ \forall x, a, h, k, w, \ \left| (\widehat{\mathbb{P}}^k - \mathbb{P}) V_{h+1}^* \Big|_{x,a,w} \right| \le 2\epsilon + \sqrt{\frac{(d \wedge S)H^2}{2N^k(x,a)} \log \frac{6H^2 SAK}{\delta\epsilon}} \right\}, \quad (G_H)$$

$$G_{\widehat{B}} := \left\{ \forall x, a, h, k, w, \ \left| (\widehat{\mathbb{P}}^k - \mathbb{P}) V_{h+1}^* \Big|_{x,a,w} \right| \le \right.$$

$$\left. 2\epsilon + \sqrt{\frac{2(d \wedge S)\left\| V_{h+1}^* \right\|_{\widehat{\mathbb{P}}^k}^2}{N^k(x,a)} \log \frac{6H^2 SAK}{\delta\epsilon}} + \frac{7(d \wedge S)H}{3N^k(x,a)} \log \frac{6H^2 SAK}{\delta\epsilon} \right\}, \quad (G_{\widehat{B}})$$

$$G_V := \left\{ \forall x, a, h, k, w, \ \left| \|V_h^*\|_{\widehat{\mathbb{P}}^k} - \|V_h^*\|_{\mathbb{P}} \Big|_{x,a,w} \right| \le 2\epsilon + \sqrt{\frac{4(d \wedge S)H^2}{N^k(x,a)} \log \frac{6H^2 SAK}{\delta\epsilon}} \right\}, \quad (G_V)$$

$$G_P := \left\{ \forall x, a, y, k, \ \left| \widehat{\mathbb{P}}^k(y \mid x, a) - \mathbb{P}(y \mid x, a) \right| \le \right.$$

$$\left. \sqrt{\frac{2\mathbb{P}(y \mid x, a)}{N^k(x,a)} \log \frac{2S^2 AK}{\delta}} + \frac{2}{3N^k(x,a)} \log \frac{2S^2 AK}{\delta} \right\}, \quad (G_P)$$

$$G_{\widehat{P}} := \left\{ \forall x, a, y, k, \ \left| \widehat{\mathbb{P}}^k(y \mid x, a) - \mathbb{P}(y \mid x, a) \right| \le \right.$$

$$\left. \sqrt{\frac{2\widehat{\mathbb{P}}^k(y \mid x, a)}{N^k(x,a)} \log \frac{2S^2 AK}{\delta}} + \frac{7}{3N^k(x,a)} \log \frac{2S^2 AK}{\delta} \right\}, \quad (G_{\widehat{P}})$$

$$G_N := \left\{ \forall x, a, k, \ N^k(x,a) \ge \frac{1}{2} \sum_{j<k} w^j(x,a) - H \log \frac{HSA}{\delta} \right\}. \quad (G_N)$$

**Lemma 1** (Probability of good events.)**.** *Each of the good event holds with probability at least $1 - \delta$. In particular, they hold simultaneously with probability at least*

$$\mathbb{P}\left\{G_H \cap G_{\widehat{B}} \cap G_V \cap G_P \cap G_{\widehat{P}} \cap G_N\right\} \geq 1 - 6\delta.$$

*Proof.* We only need to prove that each of the good event holds with probability at least $1 - \delta$.

We first show that $\mathbb{P}\left\{G_H\right\} \geq 1 - \delta$ by Hoeffding's inequality and a covering argument. In particular by Hoeffding's inequality we have that for fixed $x, a, h, k, w$,

$$\left|(\widehat{\mathbb{P}}^k - \mathbb{P})V_{h+1}^*\Big|_{x,a,w}\right| \leq \sqrt{\frac{H^2}{2N^k(x,a)} \log \frac{2}{\delta}}$$

holds with probability at least $1 - \delta$. We first apply a covering argument for the preferences set. We consider an $\frac{\epsilon}{H}$-covering $\mathcal{C}$ for the unit ball $\left\{w \in \mathbb{R}^d : \|w\|_1 \leq 1\right\}$, then $|\mathcal{C}| \leq \left(\frac{3H}{\epsilon}\right)^d$, and for any $w$ in the ball, there exists $w' \in \mathcal{C}$ such that $\|w - w'\|_1 \leq \frac{\epsilon}{H}$. Then a union bound on $w \in \mathcal{C}$ and $(x, a, k, h) \in \mathcal{S} \times \mathcal{A} \times [K] \times [H]$ yields that with probability at least $1 - \delta$, the following holds for every $(x, a, k, h, w) \in \mathcal{S} \times \mathcal{A} \times [K] \times [H] \times \mathcal{C}$:

$$\left|(\widehat{\mathbb{P}}^k - \mathbb{P})V_{h+1}^*\Big|_{x,a,w}\right| \leq \sqrt{\frac{dH^2}{2N^k(x,a)} \log \frac{6H^2SAK}{\delta\epsilon}}$$

Now consider an arbitrary $w \in \mathcal{W}$ (hence in the unit ball), let $w' \in \mathcal{C}$ be such that $\|w - w'\|_1 \leq \frac{\epsilon}{H}$, then by Lemma 2 we have that with probability at least $1 - \delta$,

$$\left|(\widehat{\mathbb{P}}^k - \mathbb{P})V_{h+1}^*\Big|_{x,a,w}\right| \leq \left|\widehat{\mathbb{P}}^k V_{h+1}^*\Big|_{x,a,w} - \widehat{\mathbb{P}}^k V_{h+1}^*\Big|_{x,a,w'}\right|$$

$$+ \left|(\widehat{\mathbb{P}}^k - \mathbb{P})V_{h+1}^*\Big|_{x,a,w'}\right| + \left|\mathbb{P}V_{h+1}^*\Big|_{x,a,w'} - \mathbb{P}V_{h+1}^*\Big|_{x,a,w}\right|$$

$$\leq 2\epsilon + \sqrt{\frac{dH^2}{2N^k(x,a)} \log \frac{6H^2SAK}{\delta\epsilon}}$$

holds for every $(x, a, k, h, w) \in \mathcal{S} \times \mathcal{A} \times [K] \times [H] \times \mathcal{W}$. Similarly, we can apply the above covering argument for the value function set by considering an $\epsilon$-covering $\mathcal{C}$ for the $H$-ball $\left\{v \in \mathbb{R}^S : \|v\|_\infty \leq H\right\}$, then $|\mathcal{C}| \leq \left(\frac{3H}{\epsilon}\right)^S$, then we obtain that with probability at least $1 - \delta$,

$$\left|(\widehat{\mathbb{P}}^k - \mathbb{P})V_{h+1}^*\Big|_{x,a,w}\right| \leq 2\epsilon + \sqrt{\frac{SH^2}{2N^k(x,a)} \log \frac{6H^2SAK}{\delta\epsilon}}$$

holds for every $x, a, k, h$ and every value functions, hence for every preference-induced value functions. The two inequalities together show that $\mathbb{P}\left\{G_H\right\} \geq 1 - \delta$.

Similarly, we can apply the covering arguments with the empirical Bernstein's inequalities [Maurer and Pontil, 2009] to show that $\mathbb{P}\left\{G_{\widehat{B}}\right\} \geq 1 - \delta$.

$\mathbb{P}\left\{G_V\right\} \geq 1 - \delta$ is proved by the covering arguments with Theorem 10 from [Maurer and Pontil, 2009].

$\mathbb{P}\left\{G_P\right\} \geq 1 - \delta$ and $\mathbb{P}\left\{G_{\widehat{P}}\right\} \geq 1 - \delta$ are proved by Bernstein and empirical Bernstein's inequalities [Maurer and Pontil, 2009], respectively.

Finally, $\mathbb{P}\left\{G_N\right\} \geq 1 - \delta$ is due to Lemma F.4 from [Dann et al., 2017].

$\square$

**Lemma 2** (Continuity)**.** *For every $h$, we have $|V_h^*(x; w) - V_h^*(x; w')| \leq (H - h + 1) \|w - w'\|_1$.*

*Proof.* We prove it by induction. For $H + 1$, we have $V_{H+1}^*(x; w) = V_{H+1}^*(x; w') = 0$. Now suppose

$$\left|V_{h+1}^*(x; w) - V_{h+1}^*(x; w')\right| \leq (H - h) \|w - w'\|_1,$$

and consider $h$. Without loss of generality, suppose $V_h^*(x; w) \geq V_h^*(x; w')$. Then

$$
\begin{aligned}
|V_h^*(x; w) - V_h^*(x; w')| &= V_h^*(x; w) - V_h^*(x; w') \\
&= \max_a Q_h^*(x, a; w) - \max_{a'} Q_h^*(x, a; w') \\
&\leq Q_h^*(x, a; w) - Q_h^*(x, a; w') \qquad (\text{let } a = \arg\max_a Q_h^*(x, a; w)) \\
&\leq |\langle w - w', \boldsymbol{r}_h(x, a) \rangle| + \sum_{y \in \mathcal{S}} \mathbb{P}(y \mid x, a) \left( V_{h+1}^*(y; w) - V_{h+1}^*(y; w') \right) \\
&\leq \|w - w'\|_1 + (H - h) \|w - w'\|_1 \\
&= (H - h + 1) \|w - w'\|_1 .
\end{aligned}
$$

Hence the conclusion holds. $\qquad\square$

**Lemma 3** (Bounds for lower order terms). *If events* $(\mathrm{G_P})$, $(\mathrm{G_{\widehat{P}}})$ *hold, then for every* $V_1, V_2$ *such that* $0 \leq V_1(x; w) \leq V_2(x; w) \leq H$, *and for every* $x, a, h, k, w$, *the following inequalities hold:*

1. $\left| (\widehat{\mathbb{P}}^k - \mathbb{P})(V_2 - V_1) \right|_{x,a,w} \leq \frac{1}{H} \left. \mathbb{P}(V_2 - V_1) \right|_{x,a,w} + \frac{2H^2 S\iota}{N^k(x,a)}$;

2. $\left| (\widehat{\mathbb{P}}^k - \mathbb{P})(V_2 - V_1) \right|_{x,a,w} \leq \frac{1}{H} \left. \widehat{\mathbb{P}}(V_2 - V_1) \right|_{x,a,w} + \frac{3H^2 S\iota}{N^k(x,a)}$;

3. $\left| (\widehat{\mathbb{P}}^k - \mathbb{P})(V_2 - V_1)^2 \right|_{x,a,w} \leq \left. \mathbb{P}(V_2 - V_1)^2 \right|_{x,a,w} + \frac{2H^2 S\iota}{N^k(x,a)}$;

4. $\left. \|V_2 - V_1\|_{\widehat{\mathbb{P}}^k}^2 \right|_{x,a,w} \leq 2 \left. \mathbb{P}(V_2 - V_1)^2 \right|_{x,a,w} + \frac{2H^2 S\iota}{N^k(x,a)}$;

*Proof.* For simplicity in this proof we denote $p(y) := \mathbb{P}(y \mid x, a)$ and $\hat{p}(y) := \widehat{\mathbb{P}}^k(y \mid x, a)$.

For the first inequality,

$$
\begin{aligned}
\left| (\widehat{\mathbb{P}}^k - \mathbb{P})(V_2 - V_1) \right|_{x,a,w} &\leq \sum_{y \in \mathcal{S}} |\hat{p}^k(y) - p(y)| \left( V_2(y; w) - V_1(y; w) \right) \\
&\leq \sum_{y \in \mathcal{S}} \left( \sqrt{\frac{2p(y)\iota}{N^k(x,a)}} + \frac{2\iota}{3N^k(x,a)} \right) (V_2(y; w) - V_1(y; w)) \qquad (\text{since } (\mathrm{G_P}) \text{ holds}) \\
&\leq \sum_{y \in \mathcal{S}} \left( \frac{p(y)}{H} + \frac{H\iota}{2N^k(x,a)} + \frac{2\iota}{3N^k(x,a)} \right) (V_2(y; w) - V_1(y; w)) \qquad (\text{use } \sqrt{ab} \leq \frac{1}{2}(a + b)) \\
&\leq \sum_{y \in \mathcal{S}} \frac{p(y)}{H} (V_2(y; w) - V_1(y; w)) + \frac{2H^2 S\iota}{N^k(x,a)} \\
&= \frac{1}{H} \left. \mathbb{P}(V_2 - V_1) \right|_{x,a,w} + \frac{2H^2 S\iota}{N^k(x,a)} .
\end{aligned}
$$

The second inequality is proved in a same way as the first one, except that in the second step we use event $(\mathrm{G_{\widehat{P}}})$ instead of $(\mathrm{G_P})$.

For the third inequality,

$$\left| (\widehat{\mathbb{P}}^k - \mathbb{P})(V_2 - V_1)^2 \Big|_{x,a,w} \right| \le \sum_y |\hat{p}(y) - p(y)| \left( V_2(y;w) - V_1(y;w) \right)^2$$

$$\le \sum_y \left( \sqrt{\frac{2p(y)\iota}{N^k(x,a)}} + \frac{2\iota}{3N^k(x,a)} \right) (V_2(y;w) - V_1(y;w))^2 \qquad \text{(by } (\mathrm{G_P}))$$

$$\le \sum_y \left( p(y) + \frac{\iota}{2N^k(x,a)} + \frac{2\iota}{3N^k(x,a)} \right) (V_2(y;w) - V_1(y;w))^2 \qquad \text{(use } \sqrt{ab} \le a/2 + b/2)$$

$$\le \mathbb{P}(V_2 - V_1)^2 \Big|_{x,a,w} + \frac{2H^2 S\iota}{N^k(x,a)}.$$

For the fourth inequality, notice the following

$$\|V_2 - V_1\|_{\widehat{\mathbb{P}}^k}^2 \le \widehat{\mathbb{P}}^k(V_2 - V_1)^2 \le \mathbb{P}(V_2 - V_1)^2 + (\widehat{\mathbb{P}}^k - \mathbb{P})(V_2 - V_1)^2,$$

and use the third inequality.

$\square$

## B.2 Proof of the Hoeffding Variant.

In this part, we follow notations in Algorithm 1 and prove the first claim in Theorem 1.

**Bonus.** We set the bonus function in Algorithm 1 to be

$$b^k(x,a) := 2\epsilon + \sqrt{\frac{(d \wedge S)H^2\iota}{2N^k(x,a)}}, \qquad \iota := \log \frac{6H^2 SAK}{\delta\epsilon} \tag{2}$$

where $\epsilon$ will be set as $\epsilon = 1/K$.

**Lemma 4** (Optimistic value function estimation). *If event* $(\mathrm{G_H})$ *holds, then* $V_h^*(x;w) \le V_h^k(x;w)$.

*Proof.* We prove the lemma by induction over $h \in [H]$. For $H+1$, $V_{H+1}^k(x;w) = V_{H+1}^*(x;w) = 0$. Now suppose $V_{h+1}^k(x;w) \ge V_{h+1}^*(x;w)$ holds for all $(x,k,w) \in \mathcal{S} \times [K] \times \mathcal{W}$, and consider $h$. Let $a = \arg\max_a Q_h^*(x,a;w)$, then by definition we have

$$V_h^*(x;w) = Q_h^*(x,a;w) = \langle w, \boldsymbol{r}_h(x,a) \rangle + \mathbb{P}V_{h+1}^* \Big|_{x,a,w},$$

$$V_h^k(x;w) \ge Q_h^k(x,a;w) = H \wedge \left( b^k(x,a) + \langle w, \boldsymbol{r}_h(x,a) \rangle + \widehat{\mathbb{P}}^k V_{h+1}^k \Big|_{x,a,w} \right).$$

If $Q_h^k(x,a;w) = H$, then $V_h^k(x;w) \ge H \ge V_h^*(x;w)$; otherwise we have

$$V_h^k(x;w) - V_h^*(x;w) \ge b^k(x,a) + \widehat{\mathbb{P}}^k V_{h+1}^k \Big|_{x,a,w} - \mathbb{P}V_{h+1}^* \Big|_{x,a;w}$$

$$\ge b^k(x,a) + \widehat{\mathbb{P}}^k V_{h+1}^* \Big|_{x,a,w} - \mathbb{P}V_{h+1}^* \Big|_{x,a;w} \qquad \text{(by induction hypothesis)}$$

$$\ge 0, \qquad \text{(by } (\mathrm{G_H}))$$

which completes the induction and finishes the proof. $\square$

**Lemma 5** (Per-episode regret decomposition). *If events* $(\mathrm{G_H})$, $(\mathrm{G_P})$ *hold, then for every* $k$, *it holds:*

$$V_1^*(x_1;w^k) - V_1^{\pi^k}(x_1;w^k) \lesssim \mathbb{E}_{\mathbb{P},\pi^k} \sum_{h=1}^H H \wedge \left( \epsilon + \sqrt{\frac{(d \wedge S)H^2\iota}{N^k(x,a)}} + \frac{H^2 S\iota}{N^k(x_h,a_h)} \right).$$

*Proof.* We first note the following recursion,

$$V_h^k(x;w) - V_h^{\pi^k}(x;w) = Q_h^k(x,a;w) - Q_h^{\pi^k}(x,a;w) \qquad \text{(set } a = \pi_h^k(x))$$

$$\leq b^k + \widehat{\mathbb{P}}^k V_{h+1}^k - \mathbb{P} V_{h+1}^{\pi^k}\Big|_{x,a,w}$$

$$= b^k + (\widehat{\mathbb{P}}^k - \mathbb{P})V_{h+1}^* + (\widehat{\mathbb{P}}^k - \mathbb{P})(V_{h+1}^k - V_{h+1}^*) + \mathbb{P}(V_{h+1}^k - V_{h+1}^{\pi^k})\Big|_{x,a,w}$$

$$\leq 2b^k + (\widehat{\mathbb{P}}^k - \mathbb{P})(V_{h+1}^k - V_{h+1}^*) + \mathbb{P}(V_{h+1}^k - V_{h+1}^{\pi^k})\Big|_{x,a,w} \qquad \text{(by } (\mathrm{G_H}))$$

$$\leq 2b^k + \frac{2H^2 S\iota}{N^k} + \left(1 + \frac{1}{H}\right)\mathbb{P}(V_{h+1}^k - V_{h+1}^{\pi^k})\Big|_{x,a,w} \qquad \text{(by Lemma 3)},$$

and by solving the recursion and noting that $(1 + 1/H)^H \leq e$ we have

$$V_1^k(x_1; w^k) - V_1^{\pi^k}(x_1; w^k) \leq e \cdot \mathbb{E}_{\mathbb{P},\pi^k} \sum_{h=1}^{H} \left(2b^k(x_h, a_h) + \frac{2H^2 S\iota}{N^k(x_h, a_h)}\right)$$

$$\lesssim \mathbb{E}_{\mathbb{P},\pi^k} \sum_{h=1}^{H} \left(\epsilon + \sqrt{\frac{(d \wedge S)H^2\iota}{N^k(x,a)}} + \frac{H^2 S\iota}{N^k(x_h, a_h)}\right).$$

The claim is proved by using Lemma 4 and noting that the value functions are bounded by $H$. $\qquad \square$

Now we are ready to prove the first part of Theorem 1.

**Theorem 6** (Restatement of Theorem 1, Hoeffding part). *Consider Algorithm 1 with bonus function (2) and $\epsilon = 1/K$, then for any sequence of the incoming preferences $\{w^1, w^2, \ldots, w^K\}$, the regret (1) of Algorithm 1 is bounded by*

$$\mathtt{regret}(K) \lesssim \sqrt{(d \wedge S) \cdot H^3 SAK \cdot \log(HSAK/\delta)} + H^2 S^2 A \log^2(HSAK/\delta)$$

*with probability at least $1 - \delta$.*

*Proof.* First by Lemma 5 we have

$$\mathtt{regret}(K) = \sum_{k=1}^{K} V_1^*(x_1; w^k) - V_1^{\pi^k}(x_1; w^k)$$

$$\lesssim \sum_{k=1}^{K} \sum_{h=1}^{H} \sum_{x,a} w_h^k(x,a) \cdot H \wedge \left(\epsilon + \frac{H^2 S\iota}{N^k(x,a)} + \sqrt{\frac{(d \wedge S)H^2\iota}{N^k(x,a)}}\right)$$

$$\lesssim \underbrace{\sum_{k=1}^{K} \sum_{h=1}^{H} \sum_{x,a\notin L^k} w_h^k(x,a) \cdot H}_{(i)} + \underbrace{\sum_{k=1}^{K} \sum_{h=1}^{H} \sum_{x,a\in L^k} w_h^k(x,a) \cdot \epsilon}_{(ii)}$$

$$+ \underbrace{\sum_{k=1}^{K} \sum_{h=1}^{H} \sum_{x,a\in L^k} w_h^k(x,a) \cdot \frac{HS\iota}{N^k(x,a)}}_{(iii)} + \underbrace{\sum_{k=1}^{K} \sum_{h=1}^{H} \sum_{x,a\in L^k} w_h^k(x,a) \cdot \sqrt{\frac{(d \wedge S)H^2\iota}{N^k(x,a)}}}_{(iv)},$$

where we set

$$L^k := \Big\{(x,a) : \sum_{j<k} w^j(x,a) \geq 2H\iota\Big\}. \tag{3}$$

We next bound each terms separately.

*Term (i).* By (3) we have that (i) $= \sum_{k=1}^{K} \sum_{x,a\notin L^k} w^k(x,a) \cdot H \lesssim H^2 SA\iota$.

*Term (ii).* By definition we have that (ii) $= \sum_{k=1}^{K} \sum_{x,a\in L^k} w^k(x,a) \cdot \epsilon \lesssim HSAK\epsilon$.

*Term (iii).* The third term is bounded as follows:

$$(iii) = \sum_{k=1}^{K} \sum_{x,a} w^k(x,a) \cdot \frac{H^2 S\iota}{N^k(x,a)} \cdot \mathbb{1}\left[\sum_{j<k} w^j(x,a) \geq 2H\iota\right] \qquad \text{(by (3))}$$

$$\lesssim \sum_{x,a} \sum_{k=1}^{K} w^k(x,a) \cdot \frac{H^2 S \iota}{\sum_{j<k} w^j(x,a) - H\iota} \cdot \mathbb{1}\left[\sum_{j<k} w^j(x,a) \geq 2H\iota\right] \qquad \text{(by } (\text{G}_{\text{N}}))$$

$$\lesssim H^2 S \iota \cdot SA \cdot \iota = H^2 S^2 A \iota^2. \qquad \text{(integration trick)}$$

*Term (iv).* The fourth term is bounded as follows:

$$\text{(iv)} = \sum_{k=1}^{K} \sum_{x,a} w^k(x,a) \cdot \sqrt{\frac{(d \wedge S)H^2\iota}{N^k(x,a)}} \cdot \mathbb{1}\left[\sum_{j<k} w^j(x,a) \geq 2H\iota\right] \qquad \text{(by (3))}$$

$$\lesssim \sum_{x,a} \sum_{k=1}^{K} w^k(x,a) \cdot \sqrt{\frac{(d \wedge S)H^2\iota}{\sum_{j<k} w^j(x,a) - H\iota}} \cdot \mathbb{1}\left[\sum_{j<k} w^j(x,a) \geq 2H\iota\right] \qquad \text{(by } (\text{G}_{\text{N}}))$$

$$\lesssim \sqrt{(d \wedge S)H^2\iota} \cdot SA \cdot \sqrt{HK} = \sqrt{(d \wedge S)H^3 SAK\iota}. \qquad \text{(integration trick)}$$

Summing up these terms, setting $\epsilon = 1/K$ and applying Lemma 1, we complete the proof.

$\square$

---

**Algorithm 4** `MO-UCBVI` (Bernstein Variant)

---

1: initialize history $\mathcal{H}^0 = \emptyset$, $\iota = \log(HSAK/\delta)$
2: **for** episode $k = 1, 2, \ldots, K$ **do**
3:    $N^k(x,a)$, $\widehat{\mathbb{P}}^k(y \mid x,a) \leftarrow$ `Empi-Prob`$(\mathcal{H}^{k-1})$
4:    receive a preference $w^k$
5:    set $V_{H+1}^k(x; w^k) = \underline{V}_{H+1}^k(x; w^k) = 0$
6:    **for** step $h = H, H-1, \ldots, 1$ **do**
7:      **for** $(x,a) \in \mathcal{S} \times \mathcal{A}$ **do**
8:         compute bonus $b_h^k(x,a) :\eqsim \sqrt{\frac{(d \wedge S)\iota}{N^k(x,a)}} \cdot \left(\left\|V_{h+1}^k\right\|_{\widehat{\mathbb{P}}^k} + \left\|V_{h+1}^k - \underline{V}_{h+1}^k\right\|_{\widehat{\mathbb{P}}^k}\right) + \frac{(d \wedge S)H\iota}{N^k(x,a)}$
9:         compute bonus $a_h^k(x,a) :\eqsim \sqrt{\frac{(d \wedge S)\iota}{N^k(x,a)}} \cdot \left(\left\|\underline{V}_{h+1}^k\right\|_{\widehat{\mathbb{P}}^k} + \left\|V_{h+1}^k - \underline{V}_{h+1}^k\right\|_{\widehat{\mathbb{P}}^k}\right) + \frac{(d \wedge S)H\iota}{N^k(x,a)}$
10:         $Q_h^k(x,a; w^k) = \min\left\{H, \langle w^k, \boldsymbol{r}_h(x,a)\rangle + b^k(x,a) + \widehat{\mathbb{P}}_h^k V_{h+1}^k\big|_{x,a,w^k}\right\}$
11:         $V_h^k(x; w^k) = \max_{a \in \mathcal{A}} Q_h^k(x,a; w^k)$
12:         $\pi_h^k(x) = \arg\max_a Q_h^k(x,a; w^k)$
13:         $\underline{Q}_h^k(x,a; w^k) = \max\left\{0, \langle w^k, \boldsymbol{r}_h(x,a)\rangle - a^k(x,a) + \widehat{\mathbb{P}}_h^k \underline{V}_{h+1}^k\big|_{x,a,w^k}\right\}$
14:         $\underline{V}_h^k(x; w^k) = \underline{Q}_h^k(x, \pi_h^k(x); w^k)$
15:      **end for**
16:    **end for**
17:    receive initial state $x_1^k = x_1$
18:    **for** step $h = 1, 2, \ldots, H$ **do**
19:      take action $a_h^k = \pi_h^k(x_h^k)$, and obtain a new state $x_{h+1}^k$
20:    **end for**
21:    update history $\mathcal{H}^k = \mathcal{H}^{k-1} \cup \{x_h^k, a_h^k\}_{h=1}^{H}$
22: **end for**

---

### B.3 Proof of Bernstein Variant

In this part, we follow notations in Algorithm 4 and prove the second claim in Theorem 1.

For a (estimated) transition kernel $\mathbb{P}$ and a (estimated) value function $V_h(y; w)$, define its one-step variance as

$$\|V_h\|_{\mathbb{P}}^2\Big|_{x,a,w} := \sum_{y \in \mathcal{S}} \mathbb{P}(y \mid x,a)\left(V_h(y; w) - \mathbb{P}V_h\big|_{x,a,w}\right)^2.$$

Moreover, we often omit $\big|_{x,a,w}$ and simply write $\|V_h\|_{\mathbb{P}}^2$ when $x, a, w$ are clear from the context.

**Bonus.** Recall that

$$a_h^k(x,a,w) := 2\epsilon + \sqrt{\frac{2(d \wedge S)\iota}{N^k(x,a)}} \cdot \left( \left\| \underline{V}_h^k \right\|_{\widehat{\mathbb{P}}^k} + \left\| V_h^k - \underline{V}_h^k \right\|_{\widehat{\mathbb{P}}^k} \right) + \frac{7(d \wedge S)H\iota}{3N^k(x,a)},$$

$$b_h^k(x,a,w) := 2\epsilon + \sqrt{\frac{2(d \wedge S)\iota}{N^k(x,a)}} \cdot \left( \left\| \overline{V}_h^k \right\|_{\widehat{\mathbb{P}}^k} + \left\| \overline{V}_h^k - \underline{V}_h^k \right\|_{\widehat{\mathbb{P}}^k} \right) + \frac{7(d \wedge S)H\iota}{3N^k(x,a)},$$
(4)

where $\epsilon$ is a parameter to be determined (we will set $\epsilon = 1/K$).

**Lemma 6** (Optimistic value estimation). *Under event* ($\mathrm{G}_{\widehat{\mathrm{B}}}$)*, we have that for every* $x,a,k,h,w$,

- $\left| (\widehat{\mathbb{P}}^k - \mathbb{P})V_{h+1}^* \Big|_{x,a,w} \right| \leq b_h^k(x,a,w) \wedge a_h^k(x,a,w)$;

- $\underline{V}_h^k(x;w) \leq V_h^*(x;w) \leq \overline{V}_h^k(x;w)$

*Proof.* We prove the claims by induction. For $H+1$, $V_{H+1}^*(x,a;w) = \overline{V}_{H+1}^k(x,a;w) = \underline{V}_{H+1}^k(x,a;w) = 0$, hence the hypotheses hold. Now suppose the hypotheses hold for $h+1$, i.e.,

$$\left| (\widehat{\mathbb{P}}^k - \mathbb{P})V_{h+1}^* \Big|_{x,a,w} \right| \leq b_h^k(x,a,w) \wedge a_h^k(x,a,w),$$
(5)

$$\underline{V}_{h+1}^k(x;w) \leq V_{h+1}^*(x;w) \leq \overline{V}_{h+1}^k(x;w)$$
(6)

and consider $h$. First we have the following two sets of inequalities:

$$\begin{aligned}
\overline{V}_h^k(x;w) - V_h^*(x;w) &\geq \overline{Q}_h^k(x,a;w) - Q_h^*(x,a;w) && (\text{set } a = \pi_h^*(x)) \\
&\geq b_h^k(x,a,w) + \widehat{\mathbb{P}}^k \overline{V}_{h+1}^k \Big|_{x,a,w} - \mathbb{P}V_{h+1}^* \Big|_{x,a,w} \\
&\geq b_h^k(x,a,w) + (\widehat{\mathbb{P}}^k - \mathbb{P})V_{h+1}^* \Big|_{x,a,w} && (\text{by } (6)) \\
&\geq 0, && (\text{by } (5))
\end{aligned}$$

and

$$\begin{aligned}
V_h^*(x;w) - \underline{V}_h^k(x;w) &\geq Q_h^*(x,a;w) - \underline{Q}_h^k(x,a;w) && (\text{set } a = \underline{\pi}_h^*(x)) \\
&\geq a_h^k(x,a,w) + \mathbb{P}V_{h+1}^* \Big|_{x,a,w} - \widehat{\mathbb{P}}^k \underline{V}_{h+1}^k \Big|_{x,a,w} \\
&\geq a_h^k(x,a,w) + (\mathbb{P} - \widehat{\mathbb{P}}^k)V_{h+1}^* \Big|_{x,a,w} && (\text{by } (6)) \\
&\geq 0. && (\text{by } (5))
\end{aligned}$$

These justify that

$$\underline{V}_h^k(x;w) \leq V_h^*(x;w) \leq \overline{V}_h^k(x;w).$$
(7)

Second, the above inequalities imply that

$$\|V_h^*\|_{\widehat{\mathbb{P}}^k} \leq \left\| \overline{V}_h^k \right\|_{\widehat{\mathbb{P}}^k} + \left\| \overline{V}_h^k - V_h^* \right\|_{\widehat{\mathbb{P}}^k} \leq \left\| \overline{V}_h^k \right\|_{\widehat{\mathbb{P}}^k} + \left\| \overline{V}_h^k - \underline{V}_h^k \right\|_{\widehat{\mathbb{P}}^k},$$
(8)

$$\|V_h^*\|_{\widehat{\mathbb{P}}^k} \leq \left\| \underline{V}_h^k \right\|_{\widehat{\mathbb{P}}^k} + \left\| V_h^* - \underline{V}_h^k \right\|_{\widehat{\mathbb{P}}^k} \leq \left\| \underline{V}_h^k \right\|_{\widehat{\mathbb{P}}^k} + \left\| \overline{V}_h^k - \underline{V}_h^k \right\|_{\widehat{\mathbb{P}}^k},$$
(9)

therefore,

$$\begin{aligned}
\left| (\widehat{\mathbb{P}}^k - \mathbb{P})V_h^* \Big|_{x,a,w} \right| &\leq 2\epsilon + \sqrt{\frac{2(d \wedge S)\iota}{N^k(x,a)}} \cdot \|V_h^*\|_{\widehat{\mathbb{P}}^k} + \frac{7(d \wedge S)H\iota}{3N^k(x,a)} && (\text{by } (\mathrm{G}_{\widehat{\mathrm{B}}})) \\
&\leq a_{h-1}^k(x,a;w) \wedge b_{h-1}^k(x,a;w). && (\text{by } (8), (9) \text{ and } (4))
\end{aligned}$$

This completes our induction. $\qquad\square$

**Lemma 7** (Bonus upper bound). *If events* ($G_{\widehat{B}}$)*,* ($G_V$) *hold, then we have that*

$$b_h^k \vee a_h^k \big|_{x,a,w} \le 3\epsilon + \sqrt{\frac{2(d \wedge S)\iota}{N^k}} \left( \left\| V_{h+1}^* \right\|_{\mathbb{P}} + 2\sqrt{\mathbb{P}(V_{h+1}^k - \underline{V}_{h+1}^k)^2} \right) + \frac{15HS\iota}{N^k} \bigg|_{x,a,w}.$$

*Proof.* We first prove the bound for $b_h^k(x,a,w)$. Note that

$$\left\| V_{h+1}^k \right\|_{\widehat{\mathbb{P}}^k} + \left\| V_{h+1}^k - \underline{V}_{h+1}^k \right\|_{\widehat{\mathbb{P}}^k} \le \left\| V_{h+1}^* \right\|_{\widehat{\mathbb{P}}^k} + \left\| V_{h+1}^k - V_{h+1}^* \right\|_{\widehat{\mathbb{P}}^k} + \left\| V_{h+1}^k - \underline{V}_{h+1}^k \right\|_{\widehat{\mathbb{P}}^k}$$

$$\le \left\| V_{h+1}^* \right\|_{\widehat{\mathbb{P}}^k} + 2 \left\| V_{h+1}^k - \underline{V}_{h+1}^k \right\|_{\widehat{\mathbb{P}}^k} \qquad \text{(by Lemma 6)}$$

$$\le 2\epsilon + \left\| V_{h+1}^* \right\|_{\mathbb{P}} + \sqrt{\frac{4(d \wedge S)H^2\iota}{N^k(x,a)}} + 2 \left\| V_{h+1}^k - \underline{V}_{h+1}^k \right\|_{\widehat{\mathbb{P}}^k} \qquad \text{(by ($G_V$))}$$

$$\le 2\epsilon + \left\| V_{h+1}^* \right\|_{\mathbb{P}} + \sqrt{\frac{4(d \wedge S)H^2\iota}{N^k}} + 2\sqrt{2\mathbb{P}(V_{h+1}^k - \underline{V}_{h+1}^k)^2 + \frac{2H^2S\iota}{N^k}} \bigg|_{x,a,w} \qquad \text{(by Lemma 3)}$$

$$\le 2\epsilon + \left\| V_{h+1}^* \right\|_{\mathbb{P}} + 2\sqrt{\mathbb{P}\left( \overline{V}_{h+1}^k - \underline{V}_{h+1}^k \right)^2} + \sqrt{\frac{50H^2S\iota}{N^k}} \bigg|_{x,a,w}.$$

Then the bound for $b_h^k(x,a,w)$ is obtained by substituting the above into (4) and use

$$\sqrt{\frac{2(d \wedge S)\iota}{N^k(x,a)}} \cdot 2\epsilon \le \epsilon^2 + \frac{2(d \wedge S)\iota}{N^k(x,a)} \le \epsilon + \frac{2HS\iota}{N^k(x,a)}.$$

Similarly we can obtain the bound for $a_h^k(x,a,w)$. $\qquad\qquad\square$

**Lemma 8** (Per-episode regret decomposition). *If events* ($G_{\widehat{B}}$)*,* ($G_V$)*,* ($G_P$) *hold, then*

$$V_1^*(x_1; w^k) - V_1^{\pi^k}(x_1; w^k) \lesssim$$

$$\mathbb{E}_{\mathbb{P},\pi^k} \sum_{h=1}^H H \wedge \left( \epsilon + \sqrt{\frac{(d \wedge S)\iota}{N^k}} \left( \left\| V_{h+1}^* \right\|_{\mathbb{P}} + \sqrt{\mathbb{P}(V_{h+1}^k - \underline{V}_{h+1}^k)^2} \right) + \frac{H^2S\iota}{N^k} \bigg|_{x_h,a_h,w^k} \right).$$

*Proof.* We first note the following recursion,

$$V_h^k(x; w) - V_h^{\pi^k}(x; w) = Q_h^k(x,a; w) - Q_h^{\pi^k}(x,a; w) \qquad \text{(set } a = \pi_h^k(x))$$

$$\le b_h^k + \widehat{\mathbb{P}}^k V_{h+1}^k - \mathbb{P}V_{h+1}^{\pi^k} \bigg|_{x,a,w}$$

$$= b_h^k + (\widehat{\mathbb{P}}^k - \mathbb{P})V_{h+1}^* + (\widehat{\mathbb{P}}^k - \mathbb{P})(V_{h+1}^k - V_{h+1}^*) + \mathbb{P}(V_{h+1}^k - V_{h+1}^{\pi^k}) \bigg|_{x,a,w}$$

$$\le 2b_h^k + (\widehat{\mathbb{P}}^k - \mathbb{P})(V_{h+1}^k - V_{h+1}^*) + \mathbb{P}(V_{h+1}^k - V_{h+1}^{\pi^k}) \bigg|_{x,a,w} \qquad \text{(by Lemma 6)}$$

$$\le 2b_h^k + \frac{2H^2S\iota}{N^k} + \left( 1 + \frac{1}{H} \right) \mathbb{P}(V_{h+1}^k - V_{h+1}^{\pi^k}) \bigg|_{x,a,w} \qquad \text{(by Lemma 3)},$$

and by solving the recursion and noting that $(1 + 1/H)^H \le e$ we have

$$V_1^k(x_1; w^k) - V_1^{\pi^k}(x_1; w^k) \le e \cdot \mathbb{E}_{\mathbb{P},\pi^k} \sum_{h=1}^H \left( 2b_h^k(x_h,a_h) + \frac{2H^2S\iota}{N^k(x_h,a_h)} \right)$$

$$\lesssim \mathbb{E}_{\mathbb{P},\pi^k} \sum_{h=1}^H \left( \epsilon + \sqrt{\frac{(d \wedge S)\iota}{N^k}} \cdot \left( \left\| V_{h+1}^* \right\|_{\mathbb{P}} + \sqrt{\mathbb{P}(V_{h+1}^k - \underline{V}_{h+1}^k)^2} \right) + \frac{H^2S\iota}{N^k} \bigg|_{x_h,a_h,w^k} \right),$$

where the last inequality is by Lemma 7. The claim is proved by using Lemma 6 and noting that the value functions are bounded by $H$.

$\qquad\qquad\square$

**Lemma 9** (Lower order regrets). *By setting $\epsilon = 1/K$, we have that*

$$\sum_{k=1}^{K} \mathbb{E}_{\mathbb{P},\pi^k} \left[ \sum_{h=1}^{H} \mathbb{P}(V_{h+1}^k - \underline{V}_{h+1}^k)^2 \Big|_{x_h,a_h,w^k} \right] \lesssim H^5 S^2 A \iota^2.$$

*Proof.* We first note the following recursion,

$$V_h^k(x;w) - \underline{V}_h^k(x;w) = Q_h^k(x,a;w) - \underline{Q}_h^k(x,a;w) \quad (\text{set } a = \pi_h^k(x))$$

$$\leq a_h^k + b_h^k + \widehat{\mathbb{P}}^k(V_{h+1}^k - \underline{V}_{h+1}^k)\Big|_{x,a,w}$$

$$= a_h^k + b_h^k + (\widehat{\mathbb{P}}^k - \mathbb{P})(V_{h+1}^k - \underline{V}_{h+1}^k) + \mathbb{P}(V_{h+1}^k - \underline{V}_{h+1}^k)\Big|_{x,a,w}$$

$$\leq a_h^k + b_h^k + \frac{2H^2 S\iota}{N^k} + \left(1 + \frac{1}{H}\right)\mathbb{P}(V_{h+1}^k - \underline{V}_{h+1}^k)\Big|_{x,a,w} \qquad (\text{by Lemma 3})$$

$$\leq 6\epsilon + 6H\sqrt{\frac{2(d\wedge S)\iota}{N^k}} + \frac{30HS\iota}{N^k} + \frac{2H^2 S\iota}{N^k} + \left(1 + \frac{1}{H}\right)\mathbb{P}(V_{h+1}^k - \underline{V}_{h+1}^k)\Big|_{x,a,w}$$

$$\qquad (\text{by Lemmas 6 and 7})$$

$$\leq 6\epsilon + \sqrt{\frac{100H^2 S\iota}{N^k}} + \frac{32H^2 S\iota}{N^k} + \left(1 + \frac{1}{H}\right)\mathbb{P}(V_{h+1}^k - \underline{V}_{h+1}^k)\Big|_{x,a,w} \quad ,$$

by solving which and noting that $(1 + 1/H)^H \leq e$ and that the value functions are bounded by $H$, we obtain that

$$V_h^k(x;w) - \underline{V}_h^k(x;w) \lesssim \mathbb{E}_{\mathbb{P},\pi^k} \sum_{t\geq h} H \wedge \left(\epsilon + \sqrt{\frac{100H^2 S\iota}{N^k(x_h,a_h)}} + \frac{H^2 S\iota}{N^k(x_h,a_h)}\right),$$

therefore

$$\left(V_h^k(x;w) - \underline{V}_h^k(x;w)\right)^2 \lesssim \left(\mathbb{E}_{\mathbb{P},\pi^k} \sum_{t\geq h} H \wedge \left(\epsilon + \sqrt{\frac{H^2 S\iota}{N^k(x_h,a_h)}} + \frac{H^2 S\iota}{N^k(x_h,a_h)}\right)\right)^2$$

$$\lesssim H \cdot \mathbb{E}_{\mathbb{P},\pi^k} \sum_{t\geq h} \left(H \wedge \left(\epsilon + \sqrt{\frac{H^2 S\iota}{N^k(x_h,a_h)}} + \frac{H^2 S\iota}{N^k(x_h,a_h)}\right)\right)^2$$

$$\lesssim H \cdot \mathbb{E}_{\mathbb{P},\pi^k} \sum_{t\geq h} H^2 \wedge \left(\epsilon^2 + \frac{H^2 S\iota}{N^k(x_h,a_h)} + \frac{H^4 S^2 \iota^2}{(N^k(x_h,a_h))^2}\right).$$

Then

$$\text{LHS} := \sum_{k=1}^{K} \mathbb{E}_{\mathbb{P},\pi^k} \sum_{h=1}^{H} \mathbb{P}(V_{h+1}^k - \underline{V}_{h+1}^k)^2\Big|_{x_h,a_h,w^k}$$

$$= \sum_{k=1}^{K}\sum_{h=1}^{H}\sum_{x,a} w_h^k(x,a)\, \mathbb{P}(V_{h+1}^k - \underline{V}_{h+1}^k)^2\Big|_{x,a,w^k}$$

$$\lesssim H \cdot \sum_{k=1}^{K}\sum_{h=1}^{H}\sum_{x,a} w_h^k(x,a) \cdot \mathbb{E}_{\mathbb{P},\pi^k} \sum_{t\geq h} H^2 \wedge \left(\epsilon^2 + \frac{H^2 S\iota}{N^k(x_h,a_h)} + \frac{H^4 S^2 \iota^2}{(N^k(x_h,a_h))^2}\right)$$

$$\lesssim H^2 \cdot \sum_{k=1}^{K}\sum_{h=1}^{H}\sum_{x,a} w_h^k(x,a) \cdot H^2 \wedge \left(\epsilon^2 + \frac{H^2 S\iota}{N^k(x_h,a_h)} + \frac{H^4 S^2 \iota^2}{(N^k(x_h,a_h))^2}\right)$$

$$\lesssim \sum_{k,h}\sum_{x,a\notin M^k} w_h^k(x,a) H^4 + \sum_{k,h}\sum_{x,a\in M^k} w_h^k(x,a)\left(H^2\epsilon^2 + \frac{H^4 S\iota}{N^k(x,a)} + \frac{H^6 S^2 \iota^2}{(N^k(x,a))^2}\right),$$

where we set
$$M^k := \{(x,a) : \sum_{j<k} w^j(x,a) \geq 2HS\iota\}.$$

Then by ($G_N$) and the integration tricks (see the proof of Theorem 7), we obtain
$$\text{LHS} \lesssim H^4 \cdot SA \cdot HS\iota + H^2\epsilon^2 \cdot SA \cdot HK + H^4 S\iota \cdot SA \cdot \iota + H^6 S^2 \iota^2 \cdot SA/(HS\iota)$$
$$\lesssim H^5 S^2 A\iota^2 + H^3 SAK\epsilon^2 \lesssim H^5 S^2 A\iota^2,$$

where the last inequality is because $\epsilon = 1/K$. $\qquad\square$

Now we are ready to prove the second part of Theorem 1.

**Theorem 7** (Restatement of Theorem 1, Bernstein part). *Consider Algorithm 4 with bonus function* (4) *and* $\epsilon = 1/K$, *then for any sequence of the incoming preferences* $\{w^1, w^2, \ldots, w^K\}$, *the regret* (1) *of Algorithm 4 is bounded by*
$$\texttt{regret}(K) \lesssim \sqrt{(d \wedge S)H^2 SAK\iota^2} + H^{2.5} S^2 A\iota^2, \quad \iota := \log(HSAK/\delta),$$
*with probability at least* $1 - \delta$.

*Proof.* First by Lemma 8 we have

$$\texttt{regret}(K) = \sum_{k=1}^{K} V_1^*(x_1; w^k) - V_1^{\pi^k}(x_1; w^k)$$

$$\lesssim \sum_{k=1}^{K} \sum_{h=1}^{H} \sum_{x,a} w_h^k(x,a) H \wedge \left( \epsilon + \frac{H^2 S\iota}{N^k} + \sqrt{\frac{(d \wedge S)\iota}{N^k}} \left( \sqrt{\mathbb{P}(V_{h+1}^k - \underline{V}_{h+1}^k)^2} + \|V_{h+1}^*\|_{\mathbb{P}} \right) \right)$$

$$\lesssim \underbrace{\sum_{k=1}^{K} \sum_{h=1}^{H} \sum_{x,a \notin L^k} w_h^k(x,a) H}_{(i)} + \underbrace{\sum_{k=1}^{K} \sum_{h=1}^{H} \sum_{x,a \in L^k} w_h^k(x,a)\epsilon}_{(ii)} + \underbrace{\sum_{k=1}^{K} \sum_{h=1}^{H} \sum_{x,a \in L^k} w_h^k(x,a) \frac{H^2 S\iota}{N^k(x,a)}}_{(iii)}$$

$$+ \underbrace{\sum_{k=1}^{K} \sum_{h=1}^{H} \sum_{x,a \in L^k} w_h^k(x,a) \cdot \sqrt{\frac{(d \wedge S)\iota}{N^k(x,a)}} \cdot \sqrt{\mathbb{P}(V_{h+1}^k - \underline{V}_{h+1}^k)^2}\Big|_{x,a,w^k}}_{(iv)}$$

$$+ \underbrace{\sum_{k=1}^{K} \sum_{h=1}^{H} \sum_{x,a \in L^k} w_h^k(x,a) \cdot \sqrt{\frac{(d \wedge S)\iota}{N^k(x,a)}} \cdot \|V_{h+1}^*\|_{\mathbb{P}}\Big|_{x,a,w^k}}_{(v)},$$

where we set
$$L^k := \{(x,a) : \sum_{j<k} w^j(x,a) \geq 2H\iota\}. \tag{10}$$

We next bound each of these terms separately.

*Term (i).* By (10) we have that (i) $= \sum_{k=1}^{K} \sum_{x,a \notin L^k} w^k(x,a) H \lesssim H^2 SA\iota$.

*Term (ii).* By definition we have that (ii) $= \sum_{k=1}^{K} \sum_{x,a \in L^k} w^k(x,a)\epsilon \lesssim HSAK\epsilon$.

*Term (iii).* The third term is bounded as follows:

$$(iii) = \sum_{k=1}^{K} \sum_{x,a} w^k(x,a) \cdot \frac{H^2 S\iota}{N^k(x,a)} \cdot \mathbb{1}\left[\sum_{j<k} w^j(x,a) \geq 2H\iota\right] \qquad \text{(by (10))}$$

$$\lesssim \sum_{x,a} \sum_{k=1}^{K} w^k(x,a) \cdot \frac{H^2 S\iota}{\sum_{j<k} w^j(x,a) - H\iota} \cdot \mathbb{1}\left[\sum_{j<k} w^j(x,a) \geq 2H\iota\right] \qquad \text{(by ($G_N$))}$$

$$\lesssim H^2 S\iota \cdot SA \cdot \iota = H^2 S^2 A\iota^2. \qquad \text{(integration trick)}$$

*Term (iv).* By Lemma 9 and the integration tricks, we bound the fourth term as follows:

$$\text{(iv)} \le \sqrt{(d \wedge S)\iota} \cdot \sqrt{\sum_{k=1}^{K} \sum_{h=1}^{H} \sum_{x,a \in L^k} \frac{w_h^k(x,a)}{N^k(x,a)}} \cdot \sqrt{\sum_{k=1}^{K} \sum_{h=1}^{H} \sum_{x,a \in L^k} w_h^k(x,a) \cdot \mathbb{P}(V_{h+1}^k - \underline{V}_{h+1}^k)^2}$$

$$\lesssim \sqrt{S\iota} \cdot \sqrt{SA\iota} \cdot \sqrt{H^5 S^2 A \iota^2} \lesssim H^{2.5} S^2 A \iota^2.$$

*Term (v).* The fifth term is the leading term. We proceed to bound this term by noting

$$\text{(v)} \le \sum_{k=1}^{K} \sum_{h=1}^{H} \sum_{x,a \in L^k} w_h^k(x,a) \cdot \left( \sqrt{\frac{(d \wedge S)\iota}{N^k(x,a)}} \cdot \left( \left\| V_{h+1}^{\pi^k} \right\|_{\mathbb{P}} + \left\| V_{h+1}^* - V_{h+1}^{\pi^k} \right\|_{\mathbb{P}} \right) \right)$$

$$\le \sqrt{(d \wedge S)\iota} \cdot \sqrt{\sum_{k=1}^{K} \sum_{h=1}^{H} \sum_{x,a \in L^k} \frac{w_h^k(x,a)}{N^k(x,a)}} \cdot$$

$$\left( \sqrt{\underbrace{\sum_{k=1}^{K} \sum_{h=1}^{H} \sum_{x,a \in L^k} w_h^k(x,a) \cdot \left\| V_{h+1}^{\pi^k} \right\|_{\mathbb{P}}^2}_{\text{(v1)}}} + \sqrt{\underbrace{\sum_{k=1}^{K} \sum_{h=1}^{H} \sum_{x,a \in L^k} w_h^k(x,a) \cdot \left\| V_{h+1}^* - V_{h+1}^{\pi^k} \right\|_{\mathbb{P}}^2}_{\text{(v2)}}} \right),$$

where

$$\text{(v1)} = \sum_{k=1}^{K} \text{Var}_{\pi^k, \mathbb{P}} \left[ \sum_{h=1}^{H} \langle w^k, \boldsymbol{r}_h(x_h, a_h) \rangle \right] \le H^2 K$$

by the law of total variance and that the cumulative reward cannot exceed $H$, and

$$\text{(v2)} \le \sum_{k=1}^{K} \sum_{h=1}^{H} \sum_{x,a \in L^k} w_h^k(x,a) \cdot \mathbb{P}(V_{h+1}^* - V_{h+1}^{\pi^k})^2 \Big|_{x,a,w^k}$$

$$\le H \cdot \sum_{k=1}^{K} \sum_{h=1}^{H} \sum_{x,a \in L^k} w_h^k(x,a) \cdot \mathbb{P}(V_{h+1}^* - V_{h+1}^{\pi^k}) \Big|_{x,a,w^k}$$

$$\le H^2 \cdot \sum_{k=1}^{K} \left( V_1^*(x_1; w^k) - V_1^{\pi^k}(x_1; w^k) \right) = H^2 \cdot \texttt{regret}(K).$$

Hence,

$$\text{(v)} \lesssim \sqrt{(d \wedge S)\iota} \cdot \sqrt{SA\iota} \cdot \left( \sqrt{H^2 K} + H \cdot \sqrt{\texttt{regret}(K)} \right)$$

$$\lesssim \sqrt{(d \wedge S)H^2 SAK\iota^2} + \sqrt{(d \wedge S)H^2 SA\iota^2} \cdot \sqrt{\texttt{regret}(K)}.$$

Summing up terms (i) to (v) and choosing $\epsilon = 1/K$, we obtain

$$\texttt{regret}(K) \lesssim H^2 SA\iota + HSAK\epsilon + H^2 S^2 A \iota^2 + H^{2.5} S^2 A \iota^2$$

$$+ \sqrt{(d \wedge S)H^2 SAK\iota^2} + \sqrt{(d \wedge S)H^2 SA\iota^2} \cdot \sqrt{\texttt{regret}(K)}$$

$$\lesssim H^{2.5} S^2 A \iota^2 + \sqrt{(d \wedge S)H^2 SAK\iota^2} + \sqrt{(d \wedge S)H^2 SA\iota^2} \cdot \sqrt{\texttt{regret}(K)},$$

solving which we obtain

$$\texttt{regret}(K) \lesssim \sqrt{(d \wedge S)H^2 SAK\iota^2} + H^{2.5} S^2 A \iota^2.$$

Applying Lemma 1 completes the proof. □

## C  Proof of Theorem 3 and Corollary 5

In this section, we follow notations in Algorithm 2 and Algorithm 3 and prove Theorem 3. Let $\overline{V}_h^k(x; w) := \max_a \overline{Q}_h^k(x, a; w)$. Let $\pi^k$ be the planning policy at the $k$-th episode and $\bar{\pi}^k$ be the exploration policy at the $k$-th episode.

**Bonus.** We set the bonus functions in Algorithm 3 and Algorithm 2 to be

$$b^k(x,a) := 2\epsilon + \sqrt{\frac{(d \wedge S)H^2\iota}{2N^k(x,a)}}, \ c^k(x,a) := \frac{3H^2S\iota}{N^k(x,a)} + 2b^k(x,a), \ \iota := \log\frac{6H^2SAK}{\delta\epsilon}, \quad (11)$$

where $\epsilon$ is a parameter to be decided later (we will set $\epsilon = 1/K$).

We note Lemma 4 applies for the planning phase. The follows lemma relates planning error with exploration regret.

**Lemma 10** (Planning error). *If events* $(G_H)$, $(G_{\widehat{P}})$ *hold, then for every* $x, a, h, k, w$,

$$V_h^k(x,w) - V_h^{\pi^k}(x,w) \leq \left(1 + \frac{1}{H}\right)^{H-h+1} \cdot \overline{V}_h^k(x).$$

*In particular* $V_1^k(x_1;w) - V_1^{\pi^k}(x_1;w) \leq e \cdot \overline{V}_1^k(x)$.

*Proof.* We prove the conclusion by induction. For $H+1$, the conclusion holds since both the left high side and the right hand side are zero. Next we assume the conclusion holds for $h+1$, i.e.,

$$V_{h+1}^k(x,w) - V_{h+1}^{\pi^k}(x,w) \leq \left(1 + \frac{1}{H}\right)^{H-h} \cdot \overline{V}_{h+1}^k(x), \quad (12)$$

and consider $h$. Recall the definitions in Algorithm 2 and Algorithm 3. Set $a = \pi_h^k(x)$, then we have

$$V_h^k(x;w) = Q_h^k(x,a;w) = H \wedge \left( \langle w, \boldsymbol{r}_h(x,a)\rangle + \widehat{\mathbb{P}}^k V_{h+1}^k \Big|_{x,a,w} + b^k(x,a) \right), \quad (13)$$

$$V_h^{\pi^k}(x;w) = Q_h^{\pi^k}(x,a;w) = \langle w, \boldsymbol{r}_h(x,a)\rangle + \mathbb{P}V_{h+1}^{\pi^k}\Big|_{x,a,w}, \quad (14)$$

and

$$\overline{V}_h^k(x) = \max_{a'} \overline{Q}_h^k(x,a') \geq \overline{Q}_h^k(x,a) = H \wedge \left( \widehat{\mathbb{P}}^k \overline{V}_{h+1}^k \Big|_{x,a} + c^k(x,a) \right).$$

If $\overline{Q}_h^k(x,a) = H$, clearly we have $V_h^k(x;w) - V_h^{\pi^k}(x;w) \leq H \leq \overline{V}_h^k(x)$, thus the conclusion holds. In the following suppose

$$\overline{V}_h^k(x) = \max_{a'} \overline{Q}_h^k(x,a') \geq \overline{Q}_h^k(x,a) = \widehat{\mathbb{P}}^k \overline{V}_{h+1}^k \Big|_{x,a} + c^k(x,a). \quad (15)$$

Then (13) and (14) yield

$$V_h^k(x,w) - V_h^{\pi^k}(x,w) \leq \widehat{\mathbb{P}}^k V_{h+1}^k \Big|_{x,a,w} - \mathbb{P}V_{h+1}^{\pi^k}\Big|_{x,a,w} + b^k(x,a)$$

$$= \widehat{\mathbb{P}}^k(V_{h+1}^k - V_{h+1}^{\pi^k}) + (\mathbb{P} - \widehat{\mathbb{P}}^k)(V_{h+1}^* - V_{h+1}^{\pi^k}) + (\widehat{\mathbb{P}}^k - \mathbb{P})V_{h+1}^* \Big|_{x,a,w} + b^k(x,w)$$

$$\leq \widehat{\mathbb{P}}^k(V_{h+1}^k - V_{h+1}^{\pi^k}) + (\mathbb{P} - \widehat{\mathbb{P}}^k)(V_{h+1}^* - V_{h+1}^{\pi^k})\Big|_{x,a,w} + 2b^k(x,a). \qquad \text{(by Lemma 4)}$$

$$\leq \left(1 + \frac{1}{H}\right) \widehat{\mathbb{P}}^k \left(V_{h+1}^k - V_{h+1}^{\pi^k}\right)\Big|_{x,a,w} + \frac{3H^2S\iota}{N^k(x,a)} + 2b^k(x,a) \qquad \text{(by Lemma 3)}$$

$$\leq \left(1 + \frac{1}{H}\right) \widehat{\mathbb{P}}^k \left(\left(1 + \frac{1}{H}\right)^{H-h} \overline{V}_{h+1}^k\right)\Big|_{x,a} + \frac{3H^2S\iota}{N^k(x,a)} + 2b^k(x,a) \qquad \text{(by (12))}$$

$$= \left(1 + \frac{1}{H}\right)^{H-h+1} \widehat{\mathbb{P}}^k \overline{V}_{h+1}^k \Big|_{x,a} + c^k(x,a) \qquad \text{(by (11))}$$

$$\leq \left(1 + \frac{1}{H}\right)^{H-h+1} \overline{Q}_h^k(x,a) \leq \left(1 + \frac{1}{H}\right)^{H-h+1} \overline{V}_h^k(x) \qquad \text{(by (15))}.$$

Thus the conclusion also holds for $h$. By induction we complete the proof. $\qquad \square$

The following lemma is a consequence of Theorem 6 with zero reward.

**Lemma 11** (Zero reward regret). *In* (11) *set* $\epsilon = 1/K$, *then with probability at least* $1 - \delta$, *the total value (regret) of Algorithm* 2 *is bounded by*

$$\sum_{k=1}^{K} \overline{V}_1^k(x_1) \lesssim \sqrt{(d \wedge S) \cdot H^3 SAK\iota} + H^2 S^2 A\iota^2, \quad \iota := \log(HSAK/\delta).$$

*Proof.* Use Theorem 6 with preference/reward set to be zero. $\square$

We are ready to prove Theorem 3.

**Theorem 8** (Restatement of Theorem 3). *Consider Algorithm* 2 *and Algorithm* 3 *with bonus function* (11) *and* $\epsilon = 1/K$, *and suppose that Algorithm* 2 *is run for*

$$K \approx \frac{(d \wedge S) \cdot H^3 SA\iota}{\epsilon^2} + \frac{H^2 S^2 A\iota^2}{\epsilon}, \quad \iota := \log\frac{HSA}{\delta\epsilon}$$

*episodes, then Algorithm* 3 *outputs an* $(\epsilon, \delta)$-*PAC policy for preference-free exploration.*

*Proof.* Recall that in the planning phase, we uniformly sample one of the $K$ policies as our final policy. Thus, with probability at least $1 - \delta$,

$$
\begin{aligned}
V_1^*(x_1; w) - V_1^\pi(x_1; w) &= \frac{1}{K} \sum_{k=1}^{K} \left( V_1^*(x_1; w) - V_1^{\pi^k}(x_1; w) \right) \\
&\leq \frac{1}{K} \sum_{k=1}^{K} \left( V_1^k(x_1; w) - V_1^{\pi^k}(x_1; w) \right) \qquad \text{(by Lemma 4)} \\
&\leq \frac{e}{K} \sum_{k=1}^{K} \overline{V}_1^k(x_1) \qquad \text{(by Lemma 10)} \\
&\lesssim \sqrt{\frac{(d \wedge S) \cdot H^3 SA\iota}{K}} + \frac{H^2 S^2 A\iota^2}{K}, \qquad \text{(by Lemma 11)}
\end{aligned}
$$

where $\iota = \log(HSAK/\delta)$, for the right hand side to be bounded by $\epsilon$, it suffices to set

$$K \approx \frac{(d \wedge S) \cdot H^3 SA}{\epsilon^2} \log\frac{HSA}{\delta\epsilon} + \frac{H^2 S^2 A}{\epsilon} \log^2 \frac{HSA}{\delta\epsilon}.$$

$\square$

## C.1 Proof of Corollary 5

For the first conclusion, we simply set $d = SA$ and apply Theorem 3.

For the second conclusion about non-stationary MDPs, the proof logic follows with small revisions.

First, as the MDP is non-stationary, we replace $\mathbb{P}$ by $\{\mathbb{P}_h\}_{h=1}^{H}$. Similar in the algorithms, we need to replace $N^k(x, a)$ by $\{N_h^k(x, a)\}_{h=1}^{H}$, which represents the number of visits to $(x, a)$ at step $h$ upto episode $k$. Then the empirical transition probability will be estimated by $\widehat{\mathbb{P}}_h^k(y \mid x, a) = N_h^k(x, a, y)/N_h^k(x, a)$, for $N_h^k(x, a) \geq 1$.

Second, note that for stationary MDP, we have $\sum_{(x,a)} N^K(x, a) = HK$, but for non-stationary MDP, this needs to be replaced by $\sum_{(x,a)} N_h^K(x, a) = K$ for every $h$. Then for non-stationary MDP, the integration tricks used in the proof of Theorem 6 needs to be revised accordingly. Let us take

term (iv) as an example, the revised analysis should be:

$$(\text{iv}) = \sum_{k=1}^{K} \sum_{h=1}^{H} \sum_{x,a} w_h^k(x,a) \cdot \sqrt{\frac{(d \wedge S)H^2\iota}{N_h^k(x,a)}} \cdot \mathbb{1}\left[\sum_{j<k} w_h^j(x,a) \geq 2\iota\right] \qquad (\text{by a revised (3)})$$

$$\lesssim \sum_{h=1}^{H} \sum_{x,a} \sum_{k=1}^{K} w_h^k(x,a) \cdot \sqrt{\frac{(d \wedge S)H^2\iota}{\sum_{j<k} w_h^j(x,a) - \iota}} \cdot \mathbb{1}\left[\sum_{j<k} w^j(x,a) \geq 2\iota\right] \quad (\text{by a revised } (\text{G}_\text{N}))$$

$$\lesssim \sqrt{(d \wedge S)H^2\iota} \cdot HSA \cdot \sqrt{K} = \sqrt{(d \wedge S)H^4SAK\iota}. \qquad (\text{integration trick})$$

Therefore the obtained bound has an enlarged $H$ dependence.

Last, repeating the proof of Theorem 3 with the revised notations and the above revised inequalities, we obtain the exploration trajectory complexity for non-stationary MDP, where the order of $H$ is enlarged due to the revised Theorem 6.

## D  Proof of Theorem 4

In this section, we follow notations in Algorithm 2 and Algorithm 3 and prove Theorem 4. Our construction of hard instance is based on the results from Jin et al. [2020].

The following two definitions are migrated from reward-free exploitation (RFE) [Jin et al., 2020] to preference-free exploitation (PFE) in MORL.

**Definition 1** (Set of MOMDPs). Fix $\mathcal{S}, \mathcal{A}, H, \boldsymbol{r}, \mathcal{W}$ as the state sets, action sets, length of horizon, reward vector, preferences set, respectively. Then a set of transition probabilities $\mathscr{P}$ induces *a set of MOMDPs*, denoted as $(\mathcal{S}, \mathcal{A}, H, \mathscr{P}, \boldsymbol{r}, \mathcal{W}) := \{(\mathcal{S}, \mathcal{A}, H, \mathbb{P}, \boldsymbol{r}, \mathcal{W}) : \mathbb{P} \in \mathscr{P}\}$.

**Definition 2** (($\epsilon, p$)-correctness). We say a PFE algorithm is ($\epsilon, p$)-correct for a set of MOMDPs $(\mathcal{S}, \mathcal{A}, H, \mathscr{P}, \boldsymbol{r}, \mathcal{W})$, if with probability at least $1 - p$,
$$V_1^{\pi_w}(x_1; w; \mathbb{P}) \geq V_1^*(x_1; w; \mathbb{P}) - \epsilon, \quad \text{for all } w \in \mathcal{W} \text{ and } \mathbb{P} \in \mathscr{P},$$
where $\pi_w$ is the policy given by the PFE algorithm for preference $w \in \mathcal{W}$.

**Basic hard instance.**  We specify the following set of MOMDPs $(\mathcal{S}, \mathcal{A}, 2, \mathscr{P}_{\texttt{basic}}(s,\epsilon), \boldsymbol{r}, \mathcal{W})$ as a basic hard instance:

- A state set $\mathcal{S} := \{s\} \cup [d]$, where $s$ is the initial state, and $|\mathcal{S}| = d + 1$.
- An action set $\mathcal{A} := [A]$.
- A horizon length $H = 2$.
- A $d$-dimensional action-independent reward vector, described by a matrix $\boldsymbol{r} := (0, I_{d \times d}) \in \mathbb{R}^{d \times (d+1)}$, i.e., the $i$-th row $\boldsymbol{r}^{(i)}(\cdot) \in \mathbb{R}^{d+1}$ sets reward 1 at state $i$, and reward 0 elsewhere.
- A preference set $\mathcal{W} := \{w \in \mathbb{R}^d, \|w\|_1 = 1\}$. For $w \in \mathcal{W}$, $w^\top \boldsymbol{r} \in \mathbb{R}^{d+1}$ gives a scalarized reward.
- A set of transition probabilities
$$\mathscr{P}_{\texttt{basic}}(s, \epsilon) := \left\{ \mathbb{P} : \forall a \in [A], \ i \in [d], \ \left|\mathbb{P}(i \,|\, s, a) - \frac{1}{d}\right| \leq \frac{\epsilon}{d}, \ \mathbb{P}(i \,|\, i, a) = 1 \right\}.$$

  For $\mathbb{P} \in \mathscr{P}_{\texttt{basic}}(s, \epsilon)$, the initial state is always set to be $s$; then the agent takes an action and transits to $[d]$ with a near uniform distribution; finally, $[d]$ are all absorbing states.

For the above basic hard instance, Jin et al. [2020] gives the following lemma to characterizes its learning complexity.

**Lemma 12** (Lower bound on the sample complexity of PFE for the basic hard instance). *Fix $\epsilon \leq 1, p \leq \frac{1}{2}, A \geq 2$, suppose $d \geq \mathcal{O}(\log A)$. There exists a distribution $\mathcal{D}$ over $\mathscr{P}_{\texttt{basic}}(s, \epsilon)$, such that any $(\epsilon/12, p)$-correct PFE algorithm* `alg` *for $(\mathcal{S}, \mathcal{A}, 2, \mathscr{P}_{\texttt{basic}}, \boldsymbol{r}, \mathcal{W})$ must satisfy*
$$\mathbb{E}_{\mathbb{P} \sim \mathcal{D}} \mathbb{E}_{\mathbb{P}, \texttt{alg}}[K] \geq \Omega\left(\frac{dA}{\epsilon^2}\right),$$
*where $K$ is the number of trajectories collected by* `alg` *in the exploration phase.*

*Proof.* See Jin et al. [2020], Lemma D.2. Note that by the construction of the basic hard instance, the preference-weighted reward recovers any reward that is action-independent, thus for the basic hard instance defined in the language of PFE, it equals to the basic hard instance studied by Jin et al. [2020], Lemma D.2 for RFE. □

**Hard instance in full version.** Based on the basic hard instance, we next build the hard instance in full version that witnesses the lower bound in Theorem 4. Let $n = 2^{\ell_0}$. The hard instance is a set of MOMDPs $(\mathcal{S}, \mathcal{A}, H, \mathscr{P}, \boldsymbol{r}, \mathcal{W})$ specified as follows:

- A state set
$$\mathcal{S} := \left\{(s, \ell) : s \in [2^\ell], \ \ell = 0, \ldots, \ell_0\right\} \bigcup \left\{(s, \ell_0 + 1) : s \in [d]\right\},$$
where $(0, 0)$ is the initial state. The states can be viewed as a $\ell_0$-layer binary tree with $d$ leaves attached to the last layer. Clearly, $|\mathcal{S}| = 2^{\ell_0+1} - 1 + d = 2n - 1 + d = \Theta(n)$.

- An action set $\mathcal{A} := [A]$.

- A horizon length $H$.

- A $2d$-dimensional action-independent reward vector, described by a matrix
$$\boldsymbol{r} := \begin{pmatrix} 0_{d \times (n-1)} & A_{d \times n} & 0_{d \times d} \\ 0_{d \times (n-1)} & 0_{d \times n} & I_{d \times d} \end{pmatrix} \in \mathbb{R}^{2d \times (2n-1+d)},$$
where $A \in \mathbb{R}^{d \times n}$ is a fixed matrix that we will specify later.

- A preference set $\mathcal{W} := \mathcal{W}_A \oplus \mathcal{W}_{\texttt{basic}} \subset \mathbb{R}^{2d}$. Here $\mathcal{W}_{\texttt{basic}} \subset \mathbb{R}^d$ is the preference set for the basic hard instance and gives the set of the last $d$-dimensions of a preference vector. And we define $\mathcal{W}_A := \{Ae_i, \ e_i \in \mathbb{R}^n, \ i = 1, \ldots, n\} \subset \mathbb{R}^d$, which gives the set of the first $d$-dimensions of a preference vector. Here $e_i$ is the $i$-th standard coordinate vector for $\mathbb{R}^n$. Then $w^\top \boldsymbol{r} \in \mathbb{R}^{2n-1+d}$ gives a scalarized reward. The first $2n - 1$ dimensions of the scalarized reward $w^\top \boldsymbol{r}$ are always zero, i.e., the reward for the states in the $0, \ldots, \ell_0$-th layers is always zero. The last $d$ dimensions of the scalarized reward is sampled from $\{A^\top Ae_i, \ i = 1, \ldots, n\}$, which specifies the reward for states in the $(\ell_0 + 1)$-th layer.

- A set of transition probabilities
$$\mathscr{P} := \big\{ \mathbb{P}(x_{\ell+1} = (s, \ell + 1) \,|\, x_\ell = (s, \ell), a = 1) = 1, \ \ell = 0, \ldots, \ell_0 - 1,$$
$$\mathbb{P}(x_{\ell+1} = (2^\ell + s, \ell + 1) \,|\, x_\ell = (s, \ell), a > 1) = 1, \ \ell = 0, \ldots, \ell_0 - 1;$$
$$\mathbb{P}(x_{\ell_0+1} \,|\, x_{\ell_0} = (s, \ell_0), a) = \mathbb{P}^s_{\texttt{basic}}, \ \mathbb{P}^s_{\texttt{basic}} \in \mathscr{P}_{\texttt{basic}}(s, \epsilon) \big\}.$$
Notice the transition probabilities up to the $\ell_0$-th layer are fixed. And the transition probability from the $\ell_0$-th layer to the $(\ell_0 + 1)$-th layer is specified by $n$ multiples of the transition probabilities that are defined earlier for the basic hard instance. In sum, a transition probability $\mathbb{P} \in \mathscr{P}$ corresponds to $n$ multiples of transition probabilities $\mathbb{P}^s_{\texttt{basic}} \in \mathscr{P}_{\texttt{basic}}(s, \epsilon)$, $s \in [n]$.

The following lemma is the key ingredient for obtaining lower bound for PFE.

**Lemma 13.** *Let $e_1, \ldots, e_n$ be the standard coordinate vector for $\mathbb{R}^n$. Let $e_0$ be the zero vector. Suppose $d \geq \frac{200}{\epsilon_1^2} \log(n + 1)$. Then there exists a matrix $A \in \mathbb{R}^{d \times n}$ such that*
$$\left\| A^\top Ae_i - e_i \right\|_\infty \leq \epsilon_1, \quad i = 1, \ldots, n.$$

*Proof.* For $d \geq \frac{8}{\epsilon^2} \log(n + 1)$, by Johnson-Lindenstrauss lemma [Johnson and Lindenstrauss, 1984], there exists matrix $A \in \mathbb{R}^{d \times n}$, such that
$$(1 - \epsilon) \|e_i - e_j\|_2 \leq \|Ae_i - Ae_j\|_2 \leq (1 + \epsilon) \|e_i - e_j\|_2, \quad \forall i, j \in \{0, 1, \ldots, n\}.$$
Thus for $i \neq j$,
$$\langle Ae_i, Ae_j \rangle = \frac{1}{2} \left( \|A(e_i - e_0)\|_2^2 + \|A(e_j - e_0)\|_2^2 - \|A(e_i - e_j)\|_2^2 \right)$$
$$\begin{cases} \leq \frac{1}{2} \left( 2(1 + \epsilon)^2 - 2(1 - \epsilon)^2 \right) = 4\epsilon, \\ \geq \frac{1}{2} \left( 2(1 - \epsilon)^2 - 2(1 + \epsilon)^2 \right) = -4\epsilon. \end{cases}$$

And for $i = j$,

$$\langle Ae_i, Ae_i \rangle = \|A(e_i - e_0)\|_2^2 \begin{cases} \leq (1+\epsilon)^2 \leq 1 + 3\epsilon, \\ \geq (1-\epsilon)^2 \geq 1 - 3\epsilon. \end{cases}$$

In sum

$$\delta_{i,j} - 4\epsilon \leq \langle A^\top Ae_i,\, e_j \rangle \leq \delta_{i,j} + 4\epsilon.$$

Note that $A^\top Ae_i = \sum_{j=1}^n \langle A^\top Ae_i, e_j \rangle$. Thus the above inequality implies $\left\| A^\top Ae_i - e_i \right\|_\infty \leq 4\epsilon$. A rescale of $\epsilon$ completes the proof. $\square$

We can then extend results from Jin et al. [2020] to PFE using Lemma 13.

**Lemma 14** (Generalized Lemma D.3 of Jin et al. [2020])**.** *Fix* $\epsilon < \frac{1}{4}$, $\epsilon_0 < \frac{1}{16H}$ *and* $\epsilon_1 < \frac{1}{16}$. *Consider* $\mathbb{P} \in \mathcal{P}$ *and* $w = (Ae_s,\, v) \in \mathcal{W}$, *where* $v \in \mathbb{R}^d$ *is nearly uniform, i.e.,*

$$\left\| v - \frac{1}{d}\mathbb{1} \right\|_\infty \leq \frac{\epsilon_0}{d}.$$

*Then if policy* $\pi$ *is* $\epsilon$-*optimal, i.e.,* $V_1^\pi(\mathbb{P}, w) \geq V_1^*(\mathbb{P}, w) - \epsilon$, *it must visit state* $(s, \ell_0)$ *with probability at least* $\frac{1}{2}$, *i.e.,*

$$\mathbb{P}^\pi[x_{\ell_0} = (s, \ell_0)] \geq \frac{1}{2}.$$

*Proof.* From the preference vector we compute the scalarized reward as $r = \left(0, A^\top Ae_s, v\right)$, i.e., states in the $0, \ldots, (\ell_0 - 1)$-th layers has zero reward, states in the $\ell_0$-th layer has reward as $A^\top Ae_s$, which takes value nearly 1 at state $(s, \ell_0)$ and value nearly 0 at other states, and states in the last layer has nearly uniform reward $v$.

Let $\bar{v} := \frac{1}{d}\sum_{y=1}^d v[y]$ be the expected reward in the final layer under a uniform visiting distribution. Then by construction we have $|v(y) - \bar{v}| \leq \epsilon_0$, which implies $|v(y) - v(y')| \leq 2\epsilon_0$. Therefore,

$$|\mathbb{P}v(\cdot) - \mathbb{P}'v(\cdot)| \leq 2\epsilon_0, \text{ for any two transition probability } \mathbb{P} \text{ and } \mathbb{P}'.$$

Now we can compute the value of a policy $\pi$:

$$
\begin{aligned}
V_1^\pi &= \langle \mathbb{P}^\pi\left[(\cdot, \ell_0)\right],\, A^\top Ae_s(\cdot)\rangle + (H - \ell_0 - 1)\langle \mathbb{P}^\pi\left[(\cdot, \ell_0 + 1)\right],\, v(\cdot)\rangle \\
&\quad \text{(since the last layer is absorbing)} \\
&\leq (1 + \epsilon_1)\mathbb{P}^\pi\left[(s, \ell_0)\right] + \epsilon_1\left(1 - \mathbb{P}^\pi\left[(s, \ell_0)\right]\right) + (H - \ell_0 - 1)\langle \mathbb{P}^\pi\left[(\cdot, \ell_0 + 1)\right],\, v(\cdot)\rangle \\
&\quad \text{(by Lemma 13)} \\
&\leq \epsilon_1 + \mathbb{P}^\pi\left[(s, \ell_0)\right] + (H - \ell_0 - 1)\langle \mathbb{P}^\pi\left[(\cdot, \ell_0 + 1)\right],\, v(\cdot)\rangle.
\end{aligned}
$$

On the other hand we can lower bound the value of the optimal policy by a policy such that $x_{\ell_0} = (s, \ell_0)$ is taken with probability 1 (this is doable since the transition probability before the $\ell_0$-th layer is deterministic):

$$V_1^* \geq A^\top Ae_s(s) + (H - \ell_0 - 1)\langle \mathbb{P}^*\left[(\cdot, \ell_0 + 1)\right],\, v(\cdot)\rangle \geq 1 - \epsilon_1 + (H - \ell_0 - 1)\langle \mathbb{P}^*\left[(\cdot, \ell_0 + 1)\right],\, v(\cdot)\rangle.$$

In sum

$$
\begin{aligned}
\frac{1}{4} \geq \epsilon &\geq V_1^* - V_1^\pi \geq 1 - \mathbb{P}^\pi\left[(s, \ell_0)\right] - 2\epsilon_1 - 2(H - \ell_0 - 1)\epsilon_0 \\
&\geq 1 - 2\epsilon_1 - 2H\epsilon_0 - \mathbb{P}^\pi\left[(s, \ell_0)\right] \geq \frac{3}{4} - \mathbb{P}^\pi\left[(s, \ell_0)\right].
\end{aligned}
$$

$\square$

**Lemma 15** (Lemma D.4 of Jin et al. [2020])**.** *Suppose* $H \geq 2(\ell_0 + 1)$. *Then a PFE algorithm* `alg` *that is* $(\epsilon, p)$-*correct for the hard instance induces* $n$ *PFE algorithms* `alg`$^s$, $s \in [n]$, *which are all* $\left(\frac{\epsilon}{4H}, p\right)$-*correct for the basic hard instance.*

*Proof.* Suppose policy $\pi$ satisfies $V_1^\pi(\mathbb{P}, w) \geq V_1^*(\mathbb{P}, w) - \epsilon$, for all $\mathbb{P} \in \mathscr{P}$ and $w \in \mathcal{W}$. Then by setting $w = (0, A^T A e_s, v)$ for $s = 1, \ldots, n$, we obtain $n$ sub-policies, each of which visits a corresponding state from $(1, \ell_0), \ldots, (n, \ell_0)$ with probability at least $\frac{1}{2}$ by Lemma 14 and induces a near-optimal policy for the basic hard instance. The last claim is since

$$
\begin{aligned}
\epsilon &\geq V_1^*(\mathbb{P}, w) - V_1^\pi(\mathbb{P}, w) \\
&\geq \mathbb{P}^\pi \left[ x_{\ell_0} = (s, \ell_0) \right] \cdot (H - \ell_0 - 1) \cdot (V_1^*(s; \mathbb{P}_{\mathtt{basic}}^s, v) - V_1^\pi(s; \mathbb{P}_{\mathtt{basic}}^s, v)) \\
&\geq \frac{1}{2}(H - \ell_0 - 1) \cdot (V_1^*(s; \mathbb{P}_{\mathtt{basic}}^s, v) - V_1^\pi(s; \mathbb{P}_{\mathtt{basic}}^s, v)) \qquad \text{(by Lemma 14)} \\
&\geq \frac{H}{4} \cdot (V_1^*(s; \mathbb{P}_{\mathtt{basic}}^s, v) - V_1^\pi(s; \mathbb{P}_{\mathtt{basic}}^s, v)), \qquad \text{(since } H \geq 2(\ell_0 + 1))
\end{aligned}
$$

which implies $V_1^\pi(s; \mathbb{P}_{\mathtt{basic}}^s, v) \geq V_1^*(s; \mathbb{P}_{\mathtt{basic}}^s, w) - \frac{\epsilon}{4H}$ holds with probability $p$.

$\square$

We are now ready to state Theorem 4 formally and deliver the proof.

**Theorem 9** (Restatement of Theorem 4). *Fix $\epsilon, p$. There exists a set of MOMDPs induced by a set of transition probabilities $\mathscr{P}$, and a distribution $\mathcal{D}$ over $\mathscr{P}$, such that if a PFE algorithm (`alg`) is $(\epsilon, p)$-correct for the set of MOMDPs, then the number of trajectories $K$ that `alg` needs to collect in the exploitation phase must satisfy*

$$
\mathbb{E}_{\mathbb{P} \sim \mathcal{D}} \mathbb{E}_{\mathbb{P}, \mathtt{alg}}[K] \geq \Omega \left( \frac{\min\{d, S\} \cdot H^2 S A}{\epsilon^2} \right).
$$

*Proof.* Let $K^s$ be the number of visits to state $(s, \ell_0)$ by `alg`. Then according to Lemma 15, there are $n$ induced algorithms $\mathtt{alg}^s, s \in [n]$ that is $(\frac{\epsilon}{4H}, p)$-correct for the basic hard instance, and each of them collects $k^s$ number of trajectories.

However, by Lemma 12 there exists a distribution $\mathcal{D}^s$ over $\mathscr{P}_{\mathtt{basic}}(s, \frac{\epsilon}{4H})$ such that

$$
\mathbb{E}_{\mathbb{P} \sim \mathcal{D}^s} \mathbb{E}_{\mathbb{P}, \mathtt{alg}^s}[K^s] \geq \Omega \left( \frac{dAH^2}{\epsilon^2} \right).
$$

Notice that during each episode, `alg` can and only can visit one of $(s, \ell_0)$ in the hard instance, thus we have $K = \sum_{s \in [n]} K^s$. Making a summation we obtain

$$
\mathbb{E}[K] \geq \Omega \left( \frac{d \cdot nAH^2}{\epsilon^2} \right) = \Omega \left( \frac{dSAH^2}{\epsilon^2} \right).
$$

On the other hand, clearly in our construction, we can set $d = n$ where Lemma 13 holds trivially. Then by the same procedure we obtain

$$
\mathbb{E}[K] \geq \Omega \left( \frac{n \cdot nAH^2}{\epsilon^2} \right) = \Omega \left( \frac{S^2 AH^2}{\epsilon^2} \right).
$$

In sum we have

$$
\mathbb{E}[K] \geq \Omega \left( \frac{\min\{d, S\} \cdot SAH^2}{\epsilon^2} \right).
$$

$\square$

# E   Proof of Theorem 2

We now prove Theorem 2 based on Theorem 9.

**Theorem 10** (Restatement of Theorem 2). *Fix $S, A, H$. Suppose $K > 0$ is sufficiently large. Then for any algorithm that runs for $K$ episodes, there exists a set of MDPs $\mathscr{P}$ and a sequence of (necessarily adversarially chosen) preferences $\{w^1, \ldots, w^K\}$, such that*

$$
\mathbb{E}_{\mathbb{P} \sim \mathscr{P}, \mathtt{alg}}[\mathtt{regret}(K)] \geq \frac{1}{c} \cdot \sqrt{\min\{d, S\} H^2 S A K}
$$

*for some absolute constant $c > 1$, where the expectation is taken with respect to the randomness of drawing an MDP and an algorithm collecting a dataset from the chosen MDP during the exploration phase.*

*Proof.* Let us fix $d, S, A, H$ and a MDP structure as described in the proof of Theorem 9. Let $K$ to be a large number, and $\epsilon := \sqrt{\min\{d, S\} H^2 SA/K_0}$ to be a small number.

We only consider the randomness of (i) choosing an MDP with transition $\mathbb{P}$ and (ii) an algorithm collecting a dataset $\mathcal{H}^K$ from the chosen MDP during the exploration phase. In order words, we will take expectation over all the randomness during the planning phase, in particular, the considered regret is understood as

$$\texttt{regret}(K) := \sum_{k=1}^{K} \mathbb{E}\left[V_1^*(x_1; w^k) - V_1^{\pi^k}(x_1; w^k) \mid \mathbb{P}, \mathcal{H}^K\right].$$

With this in mind, let us consider the following probability measures:

- Let $\boldsymbol{P}_0(\cdot)$ be the probability measure induced by the randomness of drawing a transition kernel $\mathbb{P}$ from the set of transitions $\mathscr{P}$.

- Let $\boldsymbol{P}(\cdot \mid \mathbb{P})$ to denote the conditional probability measure induced by the randomness of running an algorithm $\texttt{alg}$ on a MDP with a fixed transition kernel $\mathbb{P}$, i.e., the probability measure of collecting a dataset.

- Let $\boldsymbol{P}(\cdot)$ be the joint probability measure induced by the randomness of drawing a transition kernel $\mathbb{P}$, and the randomness of collecting a dataset.

*Converting online-MORL algorithms into PFE algorithms.* Let $\texttt{alg}$ be an MORL algorithm determined by any fixed rules (but could contain random bits), and let $\texttt{adv}$ be an (arbitrary) adversary that provides preferences for $\texttt{alg}$ in the online MORL game. Then we inductively define a preference-free exploration algorithm $\texttt{ALG}(K, \mathbb{P})$, which runs for at most $K$ episodes on an MDP with transition kernel $\mathbb{P}$:

- At episode 1, $\texttt{adv}$ chooses a preference $w^1$ based on the fixed rule of $\texttt{alg}$; and $\texttt{alg}$ interacts with $\mathbb{P}$ under the guidance of $w^1$ to collect dataset $\mathcal{H}^1$;

- At episode $k \le K$, $\texttt{adv}$ chooses a preference $w^k$ based on the fixed rule of $\texttt{alg}$ and the history $\mathcal{H}^{k-1}$; and $\texttt{alg}$ interacts with $\mathbb{P}$ under the guidance of $w^k$ and collect dataset $\mathcal{H}^k = \mathcal{H}^{k-1} \cup \{\text{new empirical observations}\}$.

- At episode $k \le K$, if the agent receives a sequence of planning preferences $w_1, \ldots, w_n$, the agent outputs a sequence of policies $\pi_1, \ldots, \pi_n$ respectively according to the planning rule that $\texttt{alg}$ adopts at the current episode (which is based on history $\mathcal{H}^{k-1}$). For simplicity, we denote $\texttt{ALG}(\mathbb{P}, k)[\cdot] : \{w\} \to \{\pi\}$ be such a planning process, i.e., $\texttt{ALG}(\mathbb{P}, k)[w_i] = \pi_i$.

We now extend $\texttt{ALG}$ into a "correct" PFE algorithm that stops at finite time:

$$\widetilde{\texttt{ALG}}(\mathbb{P}, k)[\cdot] := \begin{cases} \texttt{ALG}(\mathbb{P}, k)[\cdot], & 1 \le k \le K, \\ \texttt{PF-UCB}(\mathbb{P}, k - K)[\cdot], & K < k \le 2K, \\ \text{output random policy for any preference}, & k > 2K. \end{cases}$$

To justify the correctness of this augmented algorithm, consider two error random variables:

$$\texttt{error}(\mathbb{P}, k, \widetilde{\texttt{ALG}})[w] := \mathbb{E}[V_1^*(w; \mathbb{P}) - V_1^{\pi}(w; \mathbb{P}) \mid \mathbb{P}, \mathcal{H}^{k-1}], \quad \text{for } \pi = \widetilde{\texttt{ALG}}(\mathbb{P}, k)[w],$$

$$\texttt{error}(\mathbb{P}, k, \widetilde{\texttt{ALG}}) := \max_w \texttt{error}(\mathbb{P}, k, \widetilde{\texttt{ALG}})[w],$$

and a stopping time

$$\mathring{K} := (2K) \wedge \min\left\{k : \texttt{error}(\mathbb{P}, k, \widetilde{\texttt{ALG}}) \le \epsilon\right\}.$$

Clearly $\mathring{K} \le 2K$ is bounded.

*The augmented PFE algorithm* $\widetilde{\mathtt{ALG}}(\mathbb{P}, \mathring{K})[\cdot]$ *is correct.* By construction, we have that for any MDP with fixed transition kernel, if $\widetilde{\mathtt{ALG}}$ is run for $2K$ episodes, its output becomes the same output of $\mathtt{PF\text{-}UCB}$ that runs for $K$ episodes, which is $(\epsilon, \delta \leq 0.9)$-correct for any preference according to Theorem 8. Mathematically, for any $\mathbb{P}$,

$$\boldsymbol{P}\left\{\mathtt{error}(\mathbb{P}, 2K, \widetilde{\mathtt{ALG}}) > \epsilon \mid \mathbb{P}\right\} < \delta \leq 0.9,$$

then over the randomness of choosing a transition kernel $\mathbb{P} \sim \boldsymbol{P}_0$, we have that

$$\boldsymbol{P}\left\{\mathtt{error}(\mathbb{P}, 2K, \widetilde{\mathtt{ALG}}) > \epsilon\right\} = \mathbb{E}_{\mathbb{P} \sim \boldsymbol{P}_0}\left[\boldsymbol{P}\left\{\mathtt{error}(\mathbb{P}, 2K, \widetilde{\mathtt{ALG}}) > \epsilon \mid \mathbb{P}\right\}\right] < 0.9.$$

By discussing whether $\mathring{K} < 2K$ (where the error is smaller than $\epsilon$ with probability 1) or $\mathring{K} = 2K$ (where we apply the above inequality), we have that

$$\boldsymbol{P}\left\{\mathtt{error}(\mathbb{P}, \mathring{K}, \widetilde{\mathtt{ALG}}) > \epsilon\right\} < 0.9.$$

This implies that $\widetilde{\mathtt{ALG}}$ that runs for $\mathring{K}$ episodes is $(\epsilon, 0.9)$-correct for the set of MDP described in the proof of Theorem 9. Then by Theorem 9, we have that

$$\mathbb{E}[\mathring{K}] \geq \frac{1}{c} \cdot \frac{\min\{d, S\} H^2 SA}{\epsilon^2} = \frac{1}{c} \cdot K,$$

for some absolute constant $c > 1$. On the other hand,

$$\frac{K}{c_1} \leq \mathbb{E}[\mathring{K}] = \mathbb{E}\left[\mathbb{1}\left[\mathring{K} > \frac{K}{2c}\right] \cdot \mathring{K}\right] + \mathbb{E}\left[\mathbb{1}\left[\mathring{K} \leq \frac{K}{2c}\right] \cdot \mathring{K}\right] \leq 2K \cdot \boldsymbol{P}\left\{\mathring{K} > \frac{K}{2c}\right\} + \frac{K}{2c},$$

which implies that

$$\boldsymbol{P}\left\{\mathring{K} > \frac{K}{2c}\right\} \geq \frac{1}{4c},$$

then by the definition of $\mathring{K}$, we have that

$$\boldsymbol{P}\left\{\text{for each } 1 \leq k \leq \frac{K}{2c}, \ \mathtt{error}(\mathbb{P}, k, \widetilde{\mathtt{ALG}}) > \epsilon\right\} \geq \boldsymbol{P}\left\{\mathring{K} > \frac{K}{2c}\right\} \geq \frac{1}{4c},$$

which further yields that for each $1 \leq k \leq \frac{K}{2c_1}$,

$$\mathbb{E}\left[\mathtt{error}(\mathbb{P}, k, \widetilde{\mathtt{ALG}})\right] \geq \frac{1}{4c} \cdot \epsilon, \tag{16}$$

and

$$\mathbb{E}\left[\sum_{k=1}^{K/(2c)} \mathtt{error}(\mathbb{P}, k, \widetilde{\mathtt{ALG}})\right] \geq \frac{1}{4c} \cdot \epsilon \cdot \frac{K}{2c} = \frac{1}{8c^2} \cdot \sqrt{\min\{d, S\} H^2 SAK}. \tag{17}$$

*Regret lower bound.* Recall the definition of $\mathtt{error}(\mathbb{P}, k, \widetilde{\mathtt{ALG}})$ and note that for $k \leq K/(2c) \leq K$, $\widetilde{\mathtt{ALG}}[\mathbb{P}, k]$ is the same as $\mathtt{ALG}(\mathbb{P}, k)$, which plans in the same way as $\mathtt{alg}$ in the $k$-th episode.

We note that (17) holds for any adversary that chooses its preferences inductively based on the online MORL algorithm. Now we specify the adversary $\mathtt{adv}$ as follows: for $k \leq K/(2c)$, $\mathtt{adv}$ specifies a preference $w^{\mathtt{adv}}$ based on the rule of $\mathtt{alg}$, the dataset $\mathcal{H}^{k-1}$, and the transition kernel $\mathbb{P}$, such that

$$(w^{\mathtt{adv}})^k := \arg\max_w \mathbb{E}[V_1^*(w; \mathbb{P}) - V_1^\pi(w; \mathbb{P}) \mid \mathbb{P}, \mathcal{H}^{k-1}], \quad \text{for } \pi = \widetilde{\mathtt{ALG}}(\mathbb{P}, k)[w],$$

then

$$\mathtt{error}(\mathbb{P}, k, \widetilde{\mathtt{ALG}})[(w^{\mathtt{adv}})^k] = \mathtt{error}(\mathbb{P}, k, \widetilde{\mathtt{ALG}}).$$

As a consequence, we have that

$$\mathbb{E}[\mathtt{regret}(K)] \geq \mathbb{E}[\mathtt{regret}(K/(2c))] = \mathbb{E}\left[\sum_{k=1}^{K/(2c)} \mathtt{error}(\mathbb{P}, k, \widetilde{\mathtt{ALG}})[(w^{\mathtt{adv}})^k]\right]$$

$$= \mathbb{E}\left[\sum_{k=1}^{K/(2c)} \mathtt{error}(\mathbb{P}, k, \widetilde{\mathtt{ALG}})\right] \geq \frac{1}{8c^2} \cdot \sqrt{\min\{d, S\} H^2 SAK},$$

and a rescaling of the constant completes our proof.

$\square$

# F   Discussion on [Zhang et al., 2020a]

There is a technical error in the proof of Lemma 2 in [Zhang et al., 2020a]. In particular, the sequence considered in Page 11, between equations (16) and (17),

$$\left\{ \mathbb{1}\left[k_i \leq K\right] \cdot \left[\left[\widehat{\mathbb{P}}^{k_1} - \mathbb{P}_h\right]\left[\overline{V}_{h+1}^{\overline{\pi}_k} - V_{h+1}^{\pi_k}\right](s,a) + \left(r_h^{k_1} - \mathbb{E}[r_h](s_h^{k_i}, a_h^{k_i})\right)\right]\right\}_{i=1}^{\tau},$$

is not a martingale difference sequence, since $\pi_k$ depends on the randomness upto episode $k$, however $k_1, \ldots, k_\tau$ are all no larger than $k$. Thus Azuma-Hoeffding's inequality cannot be applied for this sequence.

To fix this error, one may consider applying a covering argument and union bound over the value functions, but the obtained bound in their Theorem 1 should be revised to $\widetilde{\mathcal{O}}\left(\log(N)H^5S^2A/\epsilon^2\right)$.