# OpenReview forum: "Accommodating Picky Customers: Regret Bound and Exploration Complexity for Multi-Objective Reinforcement Learning"
_NeurIPS.cc/2021/Conference — NeurIPS 2021 Poster_

### Official Review · Reviewer_jxvg · 2021-07-16

**Rating:** 6
**Confidence:** 4

**Summary:**

This paper investigates multi-objective reinforcement learning (MORL) in the online setting and the so-called preference-free setting, where the preference vector is given by an adversary. It proposes a model-based algorithms for each setting under tabular episodic MDPs, respectively. Both algorithms are shown to attain nearly optimal regret or sample complexity bound.

**Limitations And Societal Impact:**

Yes

**Main Review:**

MORL is an interesting and important problem in RL, and being able to tackle adversarial preference in MORL is valuable.
The algorithm design appears to follow standard value iteration algorithms in the existing literature. The theory and their proof also adapts standard techniques. The min{d, S} factor in the regret and sample complexity bounds is tight, and it sheds light on the special structure of the MORL problem.
I appreciate the discussion on the relationship between the MORL problem and other existing RL settings (adversarial, reward-free, task-agnostic etc.). The discussion brings much insight into the context of MORL.
Overall I find the paper to be clear and easy to follow.

Suggestion

On L183, Remark 2 mentions that Theorem 1 applies to general scalarization methods beyond the linear function. It would be great if the authors could mention one or two examples of such general scalarization methods, so as to make the argument more concrete.

Typo

- L101: display equation below, Q*_h(x,a;;w) has an redundant semicolon
- L244: On other the hand => On the other hand

***** Post-rebuttal ****

I have read other reviewers' comments and authors' responses. I would like to maintain my score.

**Time Spent Reviewing:**

4

---

> ### Author Response · Authors · 2021-08-10
> **Response to Reviewer jxvg**
>
> We thank you for appreciating our contributions. We will fix the typos.
>
> Our results apply to any scalarization function $r = f (\\mathbf{r}; w) $ that is (1) deterministic, (2) Lipschitz continuous for $w$, and (3) bounded between $[0,1]$ (which could be relaxed). We have not yet explored under what conditions stochastic scalarization functions can be covered, which could be an interesting future direction. We will clarify this in our next version.

---

### Official Review · Reviewer_Lo1z · 2021-07-16

**Rating:** 7
**Confidence:** 3

**Summary:**

This paper tackles the reinforcement learning problem in a multi-objective and online setting. Two different cases are considered: (i) when preferences over the multiple objectives are given at the start of each episode and (ii) when preferences are unknown and the environment needs to be explored in preparation for any arbitrary preferences. For both cases, the paper proposes optimal solutions (up to logarithmic factors, and a factor of $H$ in the second case).

**Limitations And Societal Impact:**

An important limitation that is mentioned by the authors is that the upper bound in Theorem 3 is not optimal with respect to $H$. As far as I understood, despite this limitation, their analysis in Theorem 3 still improves the existing bounds for the task-agnostic exploration setting in terms of dependence on $H$.

**Main Review:**

The paper addresses an important problem: accommodating users with varying goals and preferences, which could be relevant for many domains. Authors give the example of autonomous driving, which also explains how exactly the two settings considered in the paper might arise in a practical application. Maybe one or two additional examples from other domains could make the motivation even stronger.

Demonstrating how existing solutions for the single-objective case cannot be used in the multi-objective case when the preferences are designed adversarially (since they would suffer $O(K)$ regret) is important in motivating the need for a new algorithm like MO-UCBVI. In the current version, this demonstration is deferred to Appendix A, which does not have enough details to make a convincing argument. In particular, what "the best-in-hindsight policy" formally means and how preferences are "designed adversarially" in the experiment presented in Figure 1 needs to be explained. Considering the importance of this argument, authors might consider moving some of the discussion in Appendix A to the main paper. While I believe an empirical demonstration is sufficient for this paper, a future question might be to formally prove that single-objective solutions would indeed incur $O(K)$ regret.

In Remark 2 (line 183), it is mentioned that Theorem 1 can be extended to more general cases. Does this mean that any arbitrary deterministic function $f$ such that $r=f(\mathbf{r};w)$ can be accommodated? If not, the necessary conditions should be given formally.

Theorem 3 confirming the conjecture of Jin et al. (2020) is a nice result that makes the contributions of the paper stronger.

Corollary 5 gives a regret bound for the case when the MDP is non-stationary. Is this true for any arbitrary forms of non-stationarity? If so, how is PF-UCB able to achieve this in the setting of reward-free exploration? Why does Theorem 3 not give a similar bound for non-stationary MDPs as well (i.e. why is PF-UCB *not* able to achieve the same thing in the setting of preference-free exploration)?

While I understand the duality between preference-free and reward-free exploration quite easily, how preference-free exploration can be applied to the setting of task-agnostic exploration is not immediately clear to me (especially, why the regret bound would change from $\min\{d,S\}$ to $\log N$). Both the explanation in line 302 should be improved and a formal proof of the regret bound $\tilde{O}(\log N \cdot H^4S A/\epsilon^2)$ should be given (or an explanation of how the proof of Theorem 3 can be adapted to the setting of task-agnostic exploration).

**Time Spent Reviewing:**

4

---

> ### Author Response · Authors · 2021-08-10
> **Response to Reviewer Lo1z**
>
> Thanks for the positive evaluation of this work and the detailed comments.
>
> ### Q1: Maybe one or two additional examples from other domains could make the motivation even stronger.
> A1: Thanks for the suggestion. There are plenty of “picky-customer” examples besides the discussed autonomous driving example, to list a few: (1) medical treatment must take care of every patient even in very rare health conditions; (2) an education system should accommodate every student according to his/her own characteristics; and (3) emergency response systems have to be responsible in all extreme cases. We will strengthen our motivation with more examples in our next version.
>
> ### Q2: Demonstrating how existing solutions for the single-objective case cannot be used in the multi-objective case when the preferences are designed adversarially (since they would suffer $O(K)$ regret) is important in motivating the need for a new algorithm.
> A2: Good suggestion! We plan to move Figure 1 into Section 1 to emphasize that MORL needs new efficient algorithms. We will also add a detailed description for the experiment. In particular, the “best-in-hindsight” policy refers to a policy that is fixed across episodes and achieves maximum cumulative rewards in the whole game, i.e., $ \arg\max\_{\pi} \sum\_{k=1}^K V^{\pi}\_1(x\_1; w^k)  $.
> In the single-objective RL setting, this is the optimal solution of the underlying MDP; however in the multi-objective RL setting, this could be much worse than a time-varying policy, e.g., $ \max\_{\pi} \sum\_{k=1}^K V^{\pi}\_1(x\_1; w^k) \leq \sum\_{k=1}^K V^\*_1(x\_1; w^k) $.
>
> In experiments, the information-theoretically adversarial preferences are computationally infeasible to compute; in Figure 1 we simply use a randomly generated set of preferences, for which the “best-in-hindsight” policy already performs poorly.
>
> ### Q3: In Remark 2 (line 183), it is mentioned that Theorem 1 can be extended to more general cases. Does this mean that any arbitrary deterministic function $f$ such that $r = f ( \mathbf{r}; w)$ can be accommodated?
> A3: Our results apply to any scalarization function $r = f (\mathbf{r}; w) $ that is (1) deterministic, (2) Lipschitz continuous for $w$, and (3) bounded between $[0,1]$ (which can be relaxed). We have not yet explored under what conditions stochastic scalarization functions can be covered, which could be an interesting future direction. We will clarify this in our next version.
>
>
> ### Q4: Corollary 5 gives a regret bound for the case when the MDP is non-stationary. Is this true for any arbitrary forms of non-stationarity?
> A4: As explained in footnote 5 in page 8, the non-stationary (episodic) MDP means episodic MDP with varying (instead of fixed) transition probability at different steps (within one episode). This additional difficulty of non-stationary vs. stationary (episodic) MDP stems from the fact that: the former requires to estimate $H$ transition probabilities ${P_1, \dots, P_H}$, while the latter only requires to estimate one single $P$. This typically only adds up a certain $H$ multiplicative factor in the obtained bounds.
>
> Yes, Theorem 3 can be extended to non-stationary MDP as well. In this case the obtained bound will be updated with an additional $H$ multiplicative factor.
>
>
>
> ### Q5: Preference-free exploration can be applied to the setting of task-agnostic exploration is not immediately clear to me.
> A5: For our PFE algorithm, we can set $d = 1$ to obtain an algorithm for TAE with a single agnostic task, where we have $\min\\{d,S\\} = 1$. Then we can extend this algorithm to TAE with $N$ agnostic tasks using a union bound to have the algorithm succeed simultaneously for all $N$ tasks, which adds a $\log N$ multiplicative factor in the sample complexity bound. In this way our PFE algorithm can be converted to a TAE algorithm, and the obtained bounds have $\min\\{d,S\\}$ replaced with $\log N$.
>
> The above argument applies to any PFE algorithm that does not exploit the given reward basis $\mathbf{r}$ during exploration, which includes our PFE algorithm. We will add a proposition to formally explain this.

---

### Official Review · Reviewer_sCTs · 2021-07-16

**Rating:** 7
**Confidence:** 4

**Summary:**

This work studies the multi-objective RL (MORL) with unknown transition, where the reward function
can be parameterized with d-dimensional preference vectors. In the beginning of each episode, an adversary selects
the weight vector and reveals it to the learner, the learner then selects corresponding policy and suffers
a regret comparing with the optimal policies with respect to the given preference vector.

Besides this online MORL setting, the authors further study a preference-free exploration (PFE) setting where the learner manually separates the learning and planning phases and her performance is measured by the number of samples that are required to ensure PAC-bound. Then, the authors introduce a lower bound for PFE, which shows there is only an H factor loose.

**Limitations And Societal Impact:**

This work is pure theoretical and does not have any potential negative societal impact.

**Main Review:**

1. Contribution (Strengths)

The authors design an algorithm based on UCBVI (Azar et al., 2017) and Bernstein-style confidence intervals, which achieves sqrt(min(d,S) H^2SAK) regret. This regret guarantee is not comparable with the existing results for adversarial or stochastic episodic MDPs as the preference is known at the beginning of episode. First, the MORL provides the preference at the beginning of each episode. Second, the regret of MORL compares with a sequence of optimal policies with respect to the given preference, instead of a fixed stationary policy.

This setting can be regarded as an extension of the tabular episodic stochastic MDP setting of Azar et al. (2017). Therefore, some parts of the proof is standard and similar to the previous works such as Zanette and Brunskill (2019) and Azar et al. (2017). However, the usage of a specific covering argument and union bound in Lemma 1 is novel and interesting, which leads to a tighter regret bound with min(d,S).

Then, the authors show a lower bound for PFE based on J-L Lemma (Lemma 13) and the existing hard instance of Jin et al. (2020)., which leads to a lower bound for the online MORL setting and shows the optimality of the proposed algorithms. Though the techniques of Lemma 12-15 are not new, this lower bound is meaningful and inspiring.

2. Weaknesses

The contributions are purely theoretical, as the algorithm may be difficult to implement. Most parts of the proof are quite standard and expected.

3. Writing Issues

line 305, "Due" -> "Due to"

line 409， “Adish Singla, et al." seems wrong

line 471, "an covering" -> "a covering" (also 473, 819)

**Time Spent Reviewing:**

21

---

> ### Author Response · Authors · 2021-08-10
> **Response to Reviewer sCTs**
>
> We thank you for the positive evaluation of our work. We will make sure to fix all the typos and references.

---

### Official Review · Reviewer_5wS9 · 2021-07-17

**Rating:** 6
**Confidence:** 4

**Summary:**

 This paper studies the multi-objective reinforcement learning (MORL) problem where the reward function is an inner product of two d-dimensional vectors, an objective vector and a preference vector. Two algorithms have been proposed for known and not known preference vectors. Both algorithms are proved to achieve sub-linear regrets.

**Limitations And Societal Impact:**

yes

**Main Review:**

Strengths

1. The paper is well written and organized. The insights behind the main results are well presented.

2. This paper extends the reward-free approach recently introduced in [1] to multi-objective reinforcement learning MDPs.

Weaknesses

1. The baseline in the numerical simulations is the single-objective RL, which seems unfair because the objective functions are different. How about comparing MO-UCBVI with a more related task-agnostic algorithm [2]? Because introducing the preference is similar to changing the reward functions for different objectives.

2. Most of the theoretical analysis follows from the previous work.

3. The authors state that there is a connection between MORL and soft constrained MDPs by setting the weights to be negative. I do not quite agree. The results are derived based on the assumption that all the preference weights are bounded between [0,1]. It is not clear whether the results hold when the weights are negative.

[1]  Jin, Chi, et al. "Reward-free exploration for reinforcement learning." International Conference on Machine Learning. PMLR, 2020.

[2] Zhang, Xuezhou, and Adish Singla. "Task-agnostic exploration in reinforcement learning." arXiv preprint arXiv:2006.09497 (2020).




**Time Spent Reviewing:**

16

---

> ### Author Response · Authors · 2021-08-10
> **Response to Reviewer 5wS9**
>
> Thanks for appreciating our contributions and the paper presentation.
>
> ### Q1: The numerical simulation seems to be unfair for single-objective RL. How about comparing it to the task-agnostic algorithm?
> A1: The aim of the numerical simulation is to illustrate that existing single-objective RL algorithms cannot be directly applied in our setting.
>
> Since TAE algorithms and PFE algorithms can be converted to each other (as explained in line 302-310), we agree it makes sense to compare the empirical performance of our algorithm to a PFE version of the TAE algorithm in [2]. We will include such an experiment in the next version.
>
> ### Q2: Most of the theoretical analysis follows from the previous work.
> A2: Indeed we use some standard techniques to derive our bounds. However we would like to emphasize that both the considered MORL problem settings and results are new. It requires non-trivial efforts to establish the new algorithms. Moreover, the construction of the lower bound instances is based on a novel application of the Johnson–Lindenstrauss lemma, which is of independent interest in the field.
>
> ### Q3: The results are derived based on the assumption that all the preference weights are bounded between [0,1]. It is not clear whether the results hold when the weights are negative.
> A3: The assumption that weights are bounded in $[0,1]$ is only made for convenience. Our results naturally apply to weights bounded in $[-C, C]$, where $C$ can be any absolute positive constant. Therefore our results hold even for negative weights. We will clarify this in the next version.

---

### Official Review · Reviewer_rfjR · 2021-07-18

**Rating:** 7
**Confidence:** 4

**Summary:**

The authors propose to study an multi-objective RL in an online, episodic and tabular setting. In each episode, there is an adversarial chosen weight vector $w^k$  revealed to the agent, whose aim is to achieve a sublinear-time regret compared to an oracle who uses the optimal policy every episode. The authors propose the MO-UCBVI algorithm, which incorporate the adversarial chosen weight vectors with the learning approach by the classical UCBVI. Then the authors harness their framework on a recent proposed reward-free exploration problem, and they show that their algorithm's performance is superior to the state-of-the-art.

**Limitations And Societal Impact:**

Yes

**Main Review:**

While the paper uses technical tools from UCBVI, they involve a different definition of confident radius, by incorporating the factor $\min\{S, d\}$. Moreover, the technical contributions to the regret upper and lower bounds in the two problem settings are quite concrete, and provide strict improvement over existing results.

Two improvement to the paper are:

1) For MORL, when we specialize the multi-obj setting to the single objective setting, do we still recover the state of the art? (My understanding is that it is not, since there seem to be an extra \sqrt{S}, and the authors should check on that) While I do not think it is a major shortcoming to the paper even in the case when the best possible regret bound is not recovered, it is still informative for the authors to know how it is positioned.

2) The inclusion of numerical experiments would be informative, in particular it will shine light (empirically) on how the hardness of the problem depends on d.

Moreover, the authors should also cite the following recent work, which involves a general concave reward function for the episodic RL (NeurIPS 2020):

https://arxiv.org/pdf/2006.05051.pdf

In addition, the long version of (Cheung 2019) shows how concave reward function can model multi-objective optimization (in a different manner from the submission)

https://arxiv.org/pdf/1905.06466.pdf

Overall, I am happy with the paper's contributions.

**Time Spent Reviewing:**

4

---

> ### Author Response · Authors · 2021-08-10
> **Response to Reviewer rfjR**
>
> Thanks for the positive evaluation of our work.
>
> ### Q1: Do we recover state-of-the-art in the single-objective setting?
> A1: Yes we do. When $d = 1$ we have $\min\{d,S\} = 1$, then Theorem 1 recovers the state-of-the-art regret bound for single-objective RL. We will clarify this in our next version.
>
> ### Q2: The inclusion of numerical experiments would be informative, in particular it will shine light (empirically) on how the hardness of the problem depends on $d$.
> A2: Thanks for the suggestion. We will include experiments to illustrate the evolution of regret curves for different $d$ in our next version.
>
> ### Q3: Missing related papers.
> A3: Thanks for pointing out the related papers. We will make sure to include them in the next version.

---

### Decision · Program_Chairs · 2021-09-27

**Decision:**

Accept (Poster)

**Comment:**

This paper looks at an interesting variant of Reinforcement/Online learning, where users arrive with different preferences over several criteria. The problem is therefore linked to multi-objective optimization. Preferences could have been modelled and tackled differently than by linear aggregation, but one must start somewhere.

The content of the paper, as well as its writing, justify its acceptance.